# FasterVAR: Plug-and-Play Acceleration for Visual Autoregressive Models

**Senmao Li** [1 2 *] **Kai Wang** [3 4] **Salman Khan** [2] **Fahad Shahbaz Khan** [2 5] **Jian Yang** [1 6] **Yaxing Wang** [7]

## Abstract

Visual Autoregressive (VAR) modeling departs from the next-token prediction paradigm of traditional Autoregressive (AR) models through next-scale prediction, enabling high-quality image generation. However, the VAR paradigm suffers from sharply increased computational complexity and running time at large-scale steps. Although existing acceleration methods reduce runtime for large-scale steps, but rely on manual step selection and overlook the varying importance of different stages in the generation process. To address this challenge, we present **FasterVAR**, a systematic study and plug-and-play acceleration framework for VAR models. Our analysis shows that early steps are critical for preserving semantic and structural consistency and should remain intact, while later steps mainly refine details and can be pruned or approximated for acceleration. Building on these insights, **FasterVAR** introduces a plug-and-play acceleration strategy that exploits semantic irrelevance and low-rank properties in late-stage computations, without requiring additional training. Our proposed **FasterVAR** achieves up to 3.4× speedup with almost no performance loss, consistently outperforming existing acceleration baselines. These results highlight stage-aware design as a powerful principle for efficient visual autoregressive image generation. Code: https://github.com/sen-mao/FasterVAR

---

[*]Work done during a research intern at MBZUAI [1]PCA Lab, VCIP, College of Computer Science, Nankai University [2]Mohamed bin Zayed University of Artificial Intelligence, UAE [3]Program of Computer Science, City University of Hong Kong (Dongguan), China [4]City University of Hong Kong, HK SAR, China [5]Linkoping University, Sweden [6]PCA Lab, School of Intelligence Science and Technology, Nanjing University [7]College of Artificial Intelligence, Jilin University. Correspondence to: Kai Wang <kai.wang@cityu-dg.edu.cn>.

*Proceedings of the 43rd International Conference on Machine Learning*, Seoul, South Korea. PMLR 306, 2026. Copyright 2026 by the author(s).

## 1. Introduction

Recent developments in autoregressive (AR) models (Lee et al., 2022; Razavi et al., 2019; Sun et al., 2024; Yu et al., 2022) have yielded remarkable advances in image generation (Sun et al., 2024; Wang et al., 2025) and have been extended to serve as a unified modeling framework for both vision understanding and generation (Qu et al., 2025; Shi et al., 2026; Wu et al., 2025; Team, 2024). However, the inherent sequential nature of AR models leads to numerous decoding steps, making the inference time-consuming and costly. Departing from the sequential next-token prediction of traditional AR models, visual autoregressive (VAR) (Tian et al., 2024; Han et al., 2025) adopts a next-scale prediction paradigm, enabling more efficient for image generation.

Despite their strong generative performance, VAR models could still suffer from heavy computation and long runtime at large-scale steps. Existing methods (Guo et al., 2025; Li et al., 2025a; Chen et al., 2025a; Li et al., 2025b) accelerate VAR generation process through token reduction at large steps, but they heavily rely on manual heuristics, resulting in suboptimal acceleration. To address this challenge, we investigate how semantics and structures emerge during VAR inference to guide acceleration automatically. Our study reveals that early-scale steps in VAR inference are crucial for establishing semantics and structures, with semantics converging earlier than structures (See Fig. 1). At a specific large-scale step, the establishment of semantics and structures converges, and the remaining steps primarily perform fidelity refinement (See Fig. 1). Based on these observations, we divide the inference process into three stages: the *semantic establishment stage*, the *structure establishment stage*, and the *fidelity refinement stage* (See Fig. 1-Bottom).

Motivated by the above observations, we introduce **FasterVAR**, a plug-and-play approach that accelerates next-scale prediction VAR models without requiring additional training. We reveal that maintaining both the *semantic establishment* and *structure establishment* stages are crucial for maintaining perceptual quality, whereas the *fidelity refinement* stage can be leveraged to develop more efficient acceleration strategies. Within the fidelity refinement stage, we identify two key properties towards further acceleration: semantic irrelevance (See Fig. 2) and low-rank feature structure (See Tab. 2). Semantic irrelevance allows us to bypass

text conditioning entirely by using only a null prompt, eliminating redundant prompt computations. Meanwhile, the low-rank property enables the VAR forward pass to operate in a reduced feature space, substantially lowering inference cost. Together, these insights form the basis of **FasterVAR**'s efficient acceleration strategy.

We show that the proposed **FasterVAR** can significantly accelerate VAR image generation, achieving a **3.4×** speedup with a negligible performance drop. To summarize, our main contributions are:

- **Systematic analysis of VAR inference:** We systematically study how VAR establishes semantics and structures across scales. We show that early steps ensure semantic and structural consistency (*semantic and structure establishment stages*), while the later steps mainly refine details (*fidelity refinement stage*).

- **Plug-and-Play acceleration:** We identify two key properties, which are semantic irrelevance and low-rank properties, in the fidelity refinement stage. Leveraging these two properties, we propose the **FasterVAR** as a plug-and-play acceleration method without any additional training.

- **Extensive validation:** Experiments on the GenEval and DPG benchmarks confirm that **FasterVAR** accelerates high-quality image generation while preserving output fidelity and outperforming existing acceleration baselines. Concretely, **FasterVAR** achieves a speedup of up to 3.4× compared to baseline methods, with only minimal performance drops: a 0.01 reduction on GenEval and a 0.26 decrease on DPG.

## 2. Related work

### 2.1. Autoregressive Visual Generation

Recently, Visual Autoregressive (VAR) modeling (Tian et al., 2024; Han et al., 2025) has adopted a progressive-growing generation scheme—analogous to the mechanism first introduced in GANs (Karras et al., 2018)—to enable gradual scaling of visual generation across different resolutions. In contrast to traditional autoregressive (AR) methods (Lee et al., 2022; Sun et al., 2024; Yu et al., 2022), which adhere to a next-token prediction paradigm and thus require numerous iterative steps to produce high-quality images, VAR introduces a next-scale prediction paradigm. This shift in design allows for far more efficient synthesis of high-resolution visual content. Notably, VAR also aligns with the coarse-to-fine generation process prevalent in diffusion models (Rombach et al., 2022): it adopts this same coarse-to-fine paradigm, a choice that has contributed to its promising generative capabilities (Tian et al., 2024; Han et al., 2025; Tang et al., 2025). Despite the promising performance of VAR methods, a critical gap remains: the process by which these

models establish and refine image content during inference has not yet been systematically analyzed or studied.

### 2.2. Efficient Visual Generation

Diffusion models acceleration techniques have been extensively studied, including training-free (Ma et al., 2024a; Li et al., 2024; Tian et al., 2025; Whalen et al., 2025) and training-based methods (Luo et al., 2025; Xu et al., 2024). The exploration of efficient generation has recently drawn increasing attention in the context of autoregressive models, encompassing the traditional AR models and VAR models. For instance, SimpleAR (Wang et al., 2025) introduces a simple AR framework that achieves high-fidelity image synthesis through optimized training and inference. SJD (Teng et al., 2025) introduces a training-free probabilistic parallel decoding algorithm, enabling faster AR generation while maintaining sampling-based diversity. LANTERN (Jang et al., 2025) uses speculative decoding with a trainable drafter model to mitigate token selection ambiguity and substantially accelerate generation. Despite the acceleration, the inherent sequential nature of AR models remains a bottleneck, with high-quality image generation still taking over ten seconds.

Distinct from the traditional AR models, VAR predicts the next scale to facilitate efficient generation. However, methods for accelerating diffusion and AR models are not directly applicable to VAR, and research on VAR acceleration remains in its infancy. FastVAR (Guo et al., 2025), SparseVAR (Chen et al., 2025a) and SkipVAR (Li et al., 2025a) apply token reduction or step skipping in the manually determined large-scale steps. CoDe (Chen et al., 2025b) speeds up inference and optimizes memory usage, but it relies on collaboration between two VAR models of different scales across the inference steps. LiteVAR (Xie et al., 2024) and ScaleKV (Li et al., 2025b) improve memory efficiency by pruning attention-related tokens while with suboptimal acceleration. Recent research investigates style transfer (Park et al., 2025) and performance boosting (Chen et al.) instead of acceleration of VAR models.

In this work, we conduct a systematic analysis of how VAR models establish semantic content and structural details across diverse scale, by which we reveal the distinction of VAR generation stages. Building on the observation, we further characterize stage-dependent properties inherent in the VAR models. Finally, leveraging these properties, we propose an acceleration technique for VAR inference.

## 3. Method

We first briefly revisit VAR (Sec. 3.1), and then conduct a comprehensive analysis of the next-scale prediction process in text-to-image generation (Sec. 3.2). Our analysis

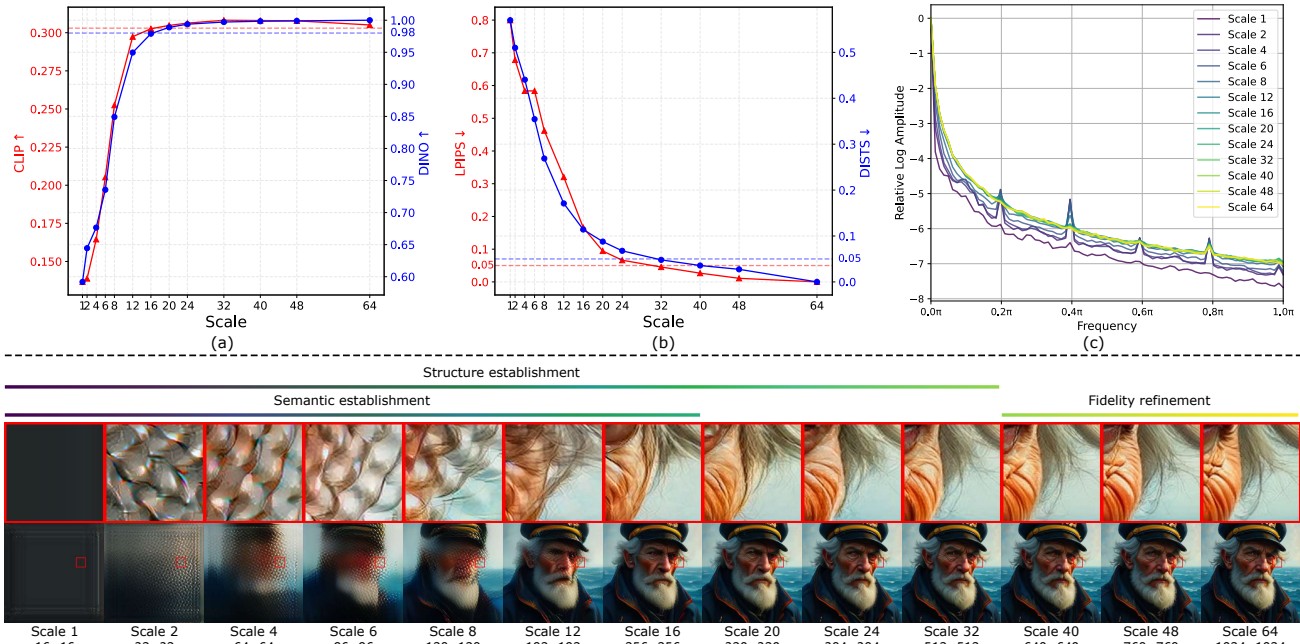

*Figure 1.* (a) Visualization of semantic evolution across all scale steps (i.e., CLIP and DINO). (b) Visualization of structure evolution on all scale steps (i.e., LPIPS and DISTS). (c) Variations of the next scale step in the frequency domain. (Bottom) Visualization of samples across all scale steps.

reveals the semantic irrelevance and low-rank properties of the VAR model. Based on these insights, we propose a novel method to accelerate VAR sampling while largely preserving generation quality and fidelity (Sec. 3.3).

### 3.1. Preliminary

Visual Autoregressive modeling (VAR) (Tian et al., 2024) redefines autoregressive modeling (AR) (Lee et al., 2022; Razavi et al., 2019; Sun et al., 2024; Yu et al., 2022) for images by shifting from next-token prediction to next-scale prediction. In this framework, each autoregressive operation generates a token map at a specific resolution scale rather than predicting individual tokens step by step. Given an continuous image feature map $\boldsymbol{F} \in \mathbb{R}^{h \times w \times d}$, VAR first quantizes it into $K$ multi-scale token maps $\boldsymbol{R} = (\boldsymbol{R}_1, \boldsymbol{R}_2, \dots, \boldsymbol{R}_K)$ with increasingly larger predefined scale $(h_k, w_k)$ for $k=1, \dots, K$. This sequence of residuals allows us to reconstruct the continuous feature $\boldsymbol{F}$ as:

$$\boldsymbol{F}_k = \sum_{i=1}^{k} \mathrm{Up}(\boldsymbol{R}_i, (h, w)), \tag{1}$$

where $\mathrm{Up}(\cdot)$ represents the upsampling operation. The multi-scale token maps $\boldsymbol{R}$ allow the decomposition of the joint probability distribution in an autoregressive manner:

$$p(\boldsymbol{R}_1, \boldsymbol{R}_2, \dots, \boldsymbol{R}_K) = \prod_{k=1}^{K} p(\boldsymbol{R}_k \mid \boldsymbol{R}_1, \boldsymbol{R}_2, \dots, \boldsymbol{R}_{k-1}), \tag{2}$$

where the initial token map $\boldsymbol{R}_1$ is derived from the text embeddings, while each subsequent $\boldsymbol{R}_k$ is generated from $\widetilde{\boldsymbol{F}}_{k-1}$, obtained via:

$$\widetilde{\boldsymbol{F}}_{k-1} = \mathrm{Down}(\boldsymbol{F}_{k-1}, (h_k, w_k)), \tag{3}$$

where $\mathrm{Down}(\cdot)$ represents the downsampling operation. $\boldsymbol{R}_k$ consists of $h_k \times w_k$ discrete tokens selected from a vocabulary of size $V$ at the *scale step* $k$. The VAR paradigm generates images in a coarse-to-fine manner with $K$ scale-up steps.

### 3.2. Observations

Here, we conduct an in-depth study of the next-scale prediction process, further exploring the semantic irrelevance and low-rank properties.

**Three-Stage Observation of Text-to-Image VAR.** As visualized in Fig. 1(Bottom), our analysis reveals three distinct generation stages of the text-to-image VAR models (Han et al., 2025). Specifically, given a pretrained VAR model, the autoregressive modeling with next-scale prediction follows the autoregressive likelihood in Eq. (2). Intuitively, at increasingly larger prediction scales, the image semantics and structures tend to be well defined.

To verify and investigate this property, we use CLIP (Hessel et al., 2021) and DINO (Oquab et al., 2023) to evaluate global and local semantics, respectively. Also, we employ LPIPS (Zhang et al., 2018) and DISTS (Ding et al., 2020) to evaluate structural consistency. The statistical results are

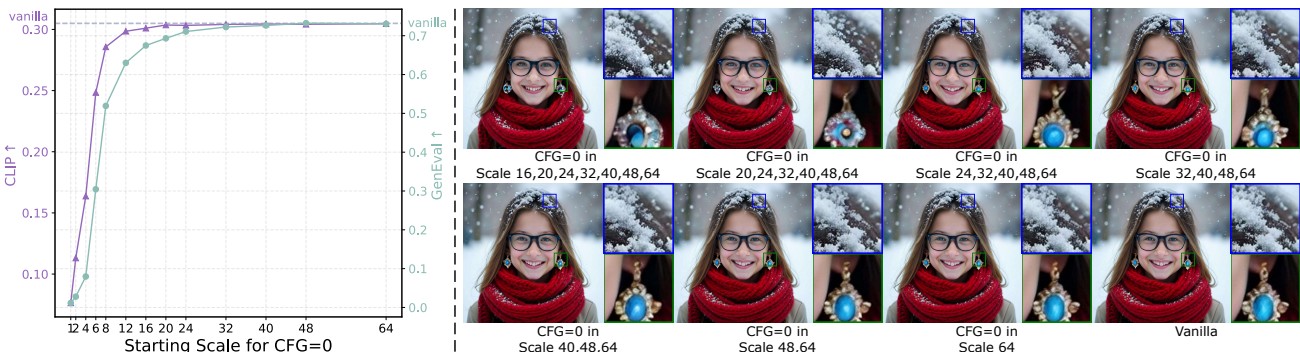

*Figure 2.* (Left) Evaluation of semantic and perceptual quality when the starting scale steps of CFG is set to 0. (Right) Sample visualizations obtained by setting CFG to 0 at large-scale steps.

shown in Fig. 1a. For both CLIP and DINO, the curves exhibit similar trends, with an initial increase followed by a stabilization towards the end. For example, at the initial scales, the value of CLIP rises quickly from around 0.13 to above 0.30, and the value of DINO increases from about 0.60 to above 0.98. This sharp improvement implies that both global and local semantics are progressively established during the early scale steps. In contrast, starting from the specific scale (i.e., 16), both the CLIP and DINO values achieve the peak plateau, indicating that the semantics have been established at these scales. Fig. 1b shows the structural evolution across different scales. Specifically, both the values of LPIPS and DISTS rapidly drop below 0.05 (at scale 32) and then level off. This indicates that the structures are progressively established during the early- and middle-scale steps. And they become well established at the large-scale steps. Moreover, we provide the frequency analysis (Fig. 1c). Both the low-frequency and high-frequency components exhibit noticeable variations during the early-scale steps, while the model almost converges in the larger scale steps.

To summarize, the early-scale steps are responsible for establishing semantics and structures, and semantics converge earlier than structures. We term these two processes as the *semantic establishment stage* and the *structure establishment stage*. In contrast, the large-scale steps mainly perform fidelity refinement, as illustrated in Fig. 1(Bottom) and we term it as the *fidelity refinement stage*. The above analyses indicate that perceptual quality is progressively established during the *semantic establishment* and *structure establishment* stages, and thus should be preserved in the original generative process. In contrast, the *fidelity refinement* stage can be exploited and operated for accelerations.

**Semantic irrelevance at large-scale steps.** Based on the aforementioned analysis, the *semantic establishment stage* has already completed the formation of image semantics. Intuitively, the large-scale steps after the *semantic establishment stage* should be unrelated to semantic generation.

To verify the intuition, we study the effect of text prompts on the semantics of the later generation stages. Recalling that VAR adopts classifier-free guidance (CFG) (Ho & Salimans, 2022) to combine text prompts with null text prompts, we set the CFG to 0 beginning at scale step $k$. Under this setup, the text prompt is omitted for steps $k$ through $K$, and the impact is evaluated using CLIP and GenEval scores. As illustrated in Fig. 2 (Left), when we set the CFG to 0 of the scale after the *semantic establishment stage* (i.e., $\{20, \ldots, 64\}$), the CLIP curve stabilizes above 0.3 and exhibits negligible changes beyond this scale step. In contrast, setting the CFG to 0 on earlier scales (i.e., $< 20$) leads to a sharp decline in the CLIP score. We further use the GenEval score to assess how well the generated images accurately reflect the intended text prompts. Both the GenEval score and the perceptual quality of image details remain close to their maximum levels (Fig. 2 (Left) and Fig. 2 (Right, 2nd row)) when $k$ is configured as during the *fidelity refinement stage* (i.e., $\{40, 48, 64\}$). In contrast, when scale is set below 40, the curve declines (Fig. 2 (Left)), accompanied by a degradation in perceptual detail quality (e.g., the "earring" in Fig. 2 (Right, 1st row)).

In conclusion, the *fidelity refinement stage* is semantics independent. Based on this observation, we are able to omit the text prompt and use only the null text prompt during the *fidelity refinement stage* to achieve the acceleration goal. This simple modification yields approximately a 1.5× speedup.

**Input Feature exhibits low-rank property.** At the intermediate $k$-th scale step of VAR, the input feature $\widetilde{\boldsymbol{F}}_{k-1}$ is obtained by interpolation, where the obtained token map of the previous scale is first upsampled (Eq. (1)) and downsampled (Eq. (3)) to match the size of the next larger scale and then fed into the model as input. This raises an important question: *Does the intermediate input feature inherently exhibit a low-rank property?*

This motivates us to use the low-rank feature to evaluate whether semantic and perceptual quality can be maintained during image generation. Specifically, for the fea-

*Table 1.* The performance in $\widetilde{F}_{k-1}$, $\widehat{F}_{k-1}$, $\widetilde{F}_r$, and $\widehat{F}_r$ with $\alpha$=0.99 and 34.4% rank. Mod. indicates the latency for the model, and Add. indicates the additional latency. ⑤/⑥ is shown in Fig. 4.

| **Inputs** | Shape | Latency↓ | | **GenEv.** | **DPG** |
|---|---|---|---|---|---|
| | | Mod. | Add. | | |
| ① $\widetilde{F}_{k-1}$ (Vanilla) | $(M, d)$ | 2.2s | 0s | 0.731 | 83.12 |
| ② $\widehat{F}_{k-1}$ | $(M, d)$ | 2.2s | 17.3s | 0.730 | 83.01 |
| ③ $\widetilde{F}_r$ w/ Eq. (4) | $(r, d)$ | 1.2s | 17.3s | 0.702 | 81.97 |
| ④ $\widetilde{F}_r$ w/o Eq. (4) | $(r, d)$ | 1.2s | 8.7s | 0.700 | 81.73 |
| ⑤ $\widehat{F}_r$ w/ Eq. (6) | $(r, d)$ | 1.2s | 0.6s | 0.690 | 81.71 |
| ⑥ $\widehat{F}_r$ w/o Eq. (6) | $(r, d)$ | 1.2s | $\gtrsim$0s | 0.720 | 82.46 |

*Vanilla and Different Low-Rank Strategies*

*Figure 3.* Visualization of VAR inference across ① vanilla, ② the low-rank feature, and ③/④ the $r$-dimensional feature.

ture at an intermediate scale step $\widetilde{F}_{k-1} \in \mathbb{R}^{M \times d}$ (i.e., $M = h_k \times w_k$, $d = 2048$ in the Infinity model (Han et al., 2025)), we perform Singular Value Decomposition (SVD) on $\widetilde{F}_{k-1} = \widetilde{U}\widetilde{\Sigma}\widetilde{V}^T$, where $\widetilde{\Sigma} = diag(\sigma_1, \cdots, \sigma_n)$, the singular values $\sigma_1 \geq \cdots \geq \sigma_n$, $n = \min(M, d)$. The cumulative energy (Jolliffe, 2025; Chong & Qu, 2025) of the top-$r$ singular values is defined as $E_r = \sum_{i=1}^{r} \sigma_i^2$, and the corresponding energy ratio is given by $\eta_r = E_r / E_n = \sum_{i=1}^{r} \sigma_i^2 / \sum_{i=1}^{n} \sigma_i^2$. A straightforward approach to constructing the low-rank feature $\widehat{F}_{k-1}$ (also shown in Fig. 3②) is to select the smallest $r$ such that

$$r = \min\{r \mid \eta_r \geq \alpha\}, \qquad (4)$$

where $\alpha \in (0, 1)$ is a threshold, as follows

$$\widetilde{F}_{k-1} \approx \widehat{F}_{k-1} = \sum_{i=1}^{r} \sigma_i u_i v_i^T \qquad (5)$$

$$= \underbrace{\left[u_1, \cdots, u_r\right]}_{\widetilde{U}_r \ (M \times r)} \underbrace{diag\{\sigma_1, \cdots, \sigma_r\}}_{\widetilde{\Sigma}_r \ (r \times r)} \underbrace{\begin{bmatrix} v_1^T \\ \vdots \\ v_r^T \end{bmatrix}}_{\widetilde{V}_r^T \ (r \times d)}$$

where $u_i$ and $v_i$ are the singular vectors in $\widetilde{U}$ and $\widetilde{V}$ corresponding to the $i$-th largest singular value $\sigma_i$. Based on the Eckart–Young–Mirsky Theorem (Eckart & Young, 1936), The low-rank feature $\widehat{F}_{k-1}$ is the most closely rank-$r$ approximation of $\widetilde{F}_{k-1}$.

We explore the impact of $\widehat{F}_{k-1}$ during the *fidelity refinement stage* on the generated image. For example, as shown in Tab. 2, we evaluate the quality of modified images against the vanilla outputs under different settings of $\alpha \in \{0.999, 0.99, 0.98, 0.97, 0.96, 0.95\}$. Setting $\alpha = \{0.999, 0.99\}$, the generated image preserves semantic and perceptual quality as the vanilla. As $\alpha$ decreases to 0.96, GenEval drops by <0.01. Achieving a slight decrease in metric values to below 0.720 with $\alpha = 0.95$ and a 14.9% rank. The results indicate that the intermediate feature exhibit a low-rank property in the *fidelity refinement stage*,

*Table 2.* Performance with the low-rank feature by varying $\alpha$.

| **Methods** | **GenEval ↑** | **DPG ↑** |
|---|---|---|
| Vanilla | 0.731 | 83.12 |
| $\alpha = 0.999$ (59.5% $rank$) | 0.729 | 83.14 |
| $\alpha = 0.99$ (34.4% $rank$) | 0.730 | 83.01 |
| $\alpha = 0.98$ (26.1% $rank$) | 0.722 | 82.81 |
| $\alpha = 0.97$ (21.1% $rank$) | 0.725 | 82.89 |
| $\alpha = 0.96$ (17.6% $rank$) | 0.726 | 82.86 |
| $\alpha = 0.95$ (14.9% $rank$) | 0.717 | 82.72 |

and this observation motivates the exploration to use the low-rank feature in this stage.

### 3.3. Plug-and-Play Acceleration for VAR

Based on all the above observations, we propose a plug-and-play acceleration method for VAR (Fig. 4 and Algorithm 1). As we state above, the VAR sampling process is conceptualized as three stages: the *semantics establishment stage*, the *structure establishment stage*, and the *fidelity refinement stage*. The semantics establishment and the structure establishment stages partially overlap in the early-scale steps, where they jointly contribute to the perceptual quality of the generated image. Therefore, we preserve the original inference process for these two stages. For the *fidelity refinement stage*, we reveal semantic irrelevance and low-rank properties, and we leverage these properties to accelerate VAR inference. The vanilla VAR inference is shown in Fig. 3①, the intermediate feature $\widetilde{F}_{k-1}$ passes through the VAR block to produce the output feature $F_k^o$, which is then quantized into $R_k$ (omitted in Fig. 3 and Fig. 4 for brevity). Our acceleration is applied at the block level (e.g., 8 blocks in the Infinity backbone).

For the low-rank property of the feature in the *fidelity refinement stage*, we can approximate the original feature of size $(M, d)$ using $r$-dimensional feature of size $(r, d)$, where $r \ll M$. Naively, as shown in Fig. 3③, for the intermediate feature $\widetilde{F}_{k-1}$ in the $k$ scale step, we first perform SVD, then determine $r$ according to Eq. (4), and construct the $r$-dimensional feature $\widetilde{F}_r = \widetilde{\Sigma}_r \widetilde{V}_r^T$ based on Eq. (5). The $r$-dimensional feature $\widetilde{F}_r$ then passes through the VAR

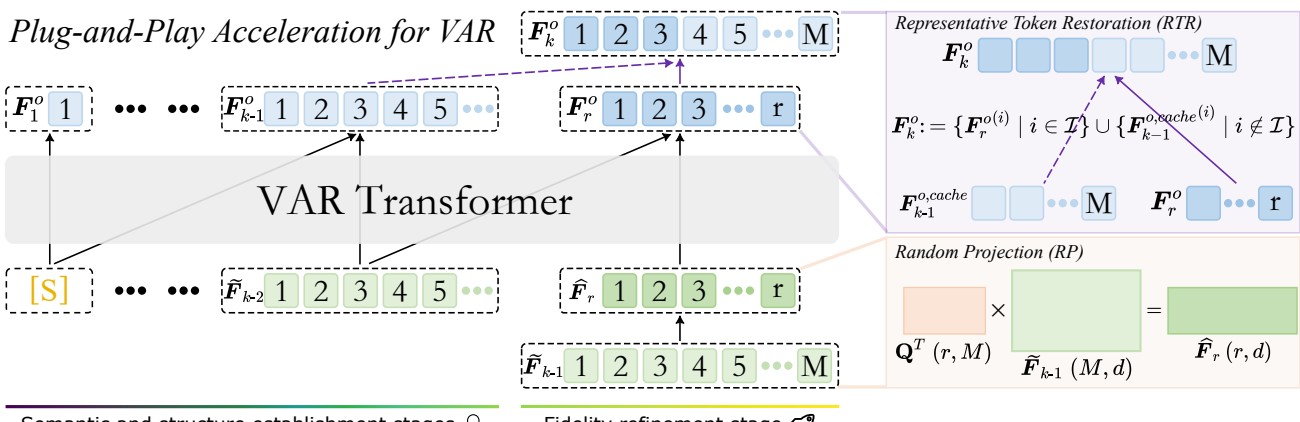

*Figure 4.* Overview of the proposed **FasterVAR** framework. We retain the original VAR inference process for the semantic and structure establishment stages, while exploiting semantic irrelevance and low-rank properties in the fidelity refinement stage to accelerate inference.

block to generate the output $\boldsymbol{F}_r^o$ (See Fig. 3③). As shown in Tab. 1③, this achieves the speedup $1.8\times$ for the VAR inference, but incurs substantial additional latency (i.e., 17.3s). The reason is that determining $r$, constructing $\widehat{\boldsymbol{F}}_r$, and constructing $\mathbf{U}_r^o$ for reconstructing the $M$-dimensional feature $\boldsymbol{F}_k^o$ all require SVD decomposition as a prerequisite, which is time-consuming. See Sec. A.2 for $\mathbf{U}_r^o$ details.

To address this issue, three aspects require considerations: **(1)** adopting an off-the-shelf value of $r$ for a given $\alpha$, **(2)** devising efficient approximation methods for $\widehat{\boldsymbol{F}}_r$ that alleviate the time-consuming computations (e.g., SVD), and **(3)** enabling the restoration of the $M$-dimensional feature $\boldsymbol{F}_k^o$. To alleviate these issues, we propose three corresponding strategies to counter as explained below.

**Predetermination Strategy.** To obtain an off-the-shelf value of $r$ given $\alpha$, we adopt a predetermined strategy based on statistical results. We find that the standard deviation is an order of magnitude smaller than the mean. That indicates for a given $\alpha$, the feature consistently exhibit similar low-rank characteristics across diverse text prompts. Specifically, given an off-the-shelf $r$, we only need to perform the $r$-rank decomposition instead of a full-rank one, reducing the decomposition time from 17.3s to 8.7s, as shown in Tab. 1③④ and Fig. 3③/④. See Sec. A.1 for details.

**Random Projection (RP) for Low-Rank Feature.** When applying the predetermined strategy, the rank of the feature is determined directly, without the requirement for either full-rank or $r$-rank decomposition. However, the $r$-dimensional feature $\widetilde{\boldsymbol{F}}_r$ needs construction. To achieve this without extra computations, we use random projection (Papadimitriou et al., 1998; Kaski, 1998; Achlioptas, 2001; Bingham & Mannila, 2001) to obtain the $r$-dimensional representation $\widehat{\boldsymbol{F}}_r$ as an approximation of $\widetilde{\boldsymbol{F}}_r$. Specifically, given the intermediate feature $\widetilde{\boldsymbol{F}}_{k-1} \in \mathbb{R}^{M \times d}$, we construct a $r$-dimensional $\widehat{\boldsymbol{F}}_r = \mathbf{Q}^T \widetilde{\boldsymbol{F}}_{k-1} \in \mathbb{R}^{r \times d}$, where

$\mathbf{Q} \in \mathbb{R}^{M \times r}$ and $\mathbf{Q}_{i,j} \sim \mathcal{N}(0, \frac{1}{r})$ (Johnson et al., 1984). After the model forward, $\widehat{\boldsymbol{F}}_r$ produces only $\boldsymbol{F}_r^o$ (See Fig. 4). Therefore, a constructed restoration matrix $\mathbf{W}_r^o$ is essential to recover the $M$-dimensional feature $\boldsymbol{F}_k^o = \mathbf{W}_r^o \boldsymbol{F}_r^o$. As the ideal restoration matrix $\mathbf{W}_r^o$ cannot be obtained directly, we instead estimate an approximation $\widehat{\mathbf{W}}_r$, which serves as the restoration matrix from $\widehat{\boldsymbol{F}}_r$ to $\widetilde{\boldsymbol{F}}_{k-1}$.

Then, to recover the intermediate feature $\widetilde{\boldsymbol{F}}_{k-1}$, we estimate the restoration matrix $\widehat{\mathbf{W}}_r \in \mathbb{R}^{M \times r}$ by solving the following linear least-squares (LLS) problem

$$\min_{\widehat{\mathbf{W}}_r} \|\widehat{\boldsymbol{F}}_r^T \widehat{\mathbf{W}}_r^T - \widetilde{\boldsymbol{F}}_{k-1}^T\|_F, \qquad (6)$$

where $\|\cdot\|_F$ denotes the Frobenius norm. As shown in Tab. 1⑤, using $\widehat{\boldsymbol{F}}_r$ for the VAR inference achieves the speedup $1.8\times$ with less additional latency (i.e., 0.6s).

**Representative Token Restoration (RTR).** Constructing $\mathbf{W}_r^o$ via an approximation $\widehat{\mathbf{W}}_r$ introduces extra latency from solving the LLS problem (i.e., Eq. (6)). To avoid solving the LLS problem to obtain $\mathbf{W}_r^o$, we regarded $\boldsymbol{F}_r^o$ as the $r$ representative feature of $\boldsymbol{F}_k^o$ according to the indices $\mathcal{I}$, while the remaining tokens are filled with the cached $\boldsymbol{F}_{k-1}^o$, as in (Guo et al., 2025). Here, since $\boldsymbol{F}_k^o$ is unavailable in advance, we instead sample $r$ rows from the corresponding input feature $\widetilde{\boldsymbol{F}}_{k-1}$ and denote their indices as $\mathcal{I}$, following (Frieze et al., 2004). See Sec. A.3 for $\mathbf{W}_r^o$ details.

Specifically, as shown in Fig. 4, given the cached counterpart feature $\boldsymbol{F}_{k-1}^o$ in the $(k-1)$-th scale, we first upsample it to match the dimension of $\boldsymbol{F}_k^o$

$$\boldsymbol{F}_{k-1}^{o,cache} = \mathrm{Up}(\boldsymbol{F}_{k-1}^o). \qquad (7)$$

Then, based on the assumption in (Frieze et al., 2004), we sample the most important $r$ rows from $\widetilde{\boldsymbol{F}}_{k-1}$, with each row $i$ chosen with probability $P_i \geq \|\widetilde{\boldsymbol{F}}_{k-1}^{(i)}\|^2 \big/ \|\widetilde{\boldsymbol{F}}_{k-1}\|_F^2$,

*Table 3.* Quantitative comparisons of perceptual quality on the GenEval and DPG Benchmarks.

| Methods | #Speed↑ | #Latency↓ | #Param↓ | GenEval↑ | | | | DPG↑ | | |
|---|---|---|---|---|---|---|---|---|---|---|
| | | | | Two Obj. | Position | Color Attri. | **Overall** | Global | Relation | **Overall** |
| SDXL (Podell et al., 2024) | - | 4.3s | 2.6B | 0.74 | 0.15 | 0.23 | 0.55 | 83.27 | 86.76 | 74.65 |
| LlamaGen (Sun et al., 2024) | - | 37.7s | 0.8B | 0.34 | 0.07 | 0.04 | 0.32 | - | - | 65.16 |
| Show-o (Xie et al., 2025) | - | 50.3s | 1.3B | 0.80 | 0.31 | 0.50 | 0.68 | - | - | 67.48 |
| PixArt-Sigma (Chen et al., 2024) | - | 2.7s | 0.6B | 0.62 | 0.14 | 0.27 | 0.55 | 86.89 | 86.59 | 80.54 |
| HART (Tang et al., 2025) | - | 0.95s | 0.7B | 0.62 | 0.13 | 0.18 | 0.51 | - | - | 80.89 |
| DALL-E 3 (Betker et al., 2023) | - | - | - | - | - | - | 0.67 | 90.97 | 90.58 | 83.50 |
| Emu3 (Wang et al., 2024) | - | - | 8.5B | 0.81 | 0.49 | 0.45 | 0.66 | - | - | 81.60 |
| Infinity-2B (Han et al., 2025) | 1.0× | 2.2s | 2.0B | 0.85 | 0.45 | 0.54 | 0.73 | 85.10 | 92.37 | 83.12 |
| FastVAR (Guo et al., 2025) | 2.75× | 0.80s | 2.0B | 0.81 | 0.45 | 0.52 | 0.72 | 85.41 | 92.76 | 82.86 |
| SkipVAR (Li et al., 2025a) | 2.62× | - | 2.0B | 0.84 | 0.39 | 0.60 | 0.72 | 84.19 | 93.15 | 83.16 |
| **Ours** | 3.4× | 0.64s | 2.0B | 0.84 | 0.43 | 0.56 | 0.72 | 82.67 | 93.50 | 82.86 |
| Infinity-8B (Han et al., 2025) | 1.0× | 4.80s | 8.0B | 0.90 | 0.62 | 0.67 | 0.79 | 85.10 | 94.50 | 86.60 |
| **Ours** | 2.7× | 1.77s | 8.0B | 0.87 | 0.60 | 0.66 | 0.78 | 85.71 | 94.43 | 86.05 |
| STAR (Ma et al., 2024b) | 1.0× | 2.0s | 1.7B | 0.54 | 0.09 | 0.08 | 0.51 | - | - | - |
| **Ours** | 1.74× | 1.15s | 1.7B | 0.54 | 0.08 | 0.09 | 0.51 | - | - | - |

and record their indices of the chosen rows as $\mathcal{I}$. Thus, we can redefine $\boldsymbol{F}_k^o$ as

$$\boldsymbol{F}_k^o := \{\boldsymbol{F}_r^{o(i)} \mid i \in \mathcal{I}\} \cup \{\boldsymbol{F}_{k-1}^{o,cache(i)} \mid i \notin \mathcal{I}\} \quad (8)$$

Formally, $\boldsymbol{F}_r^{o(i)}$ denotes that, for $i \in \mathcal{I}$, the $i$-th row of $\boldsymbol{F}_k^o$ is taken from the corresponding token of $\boldsymbol{F}_r^o$, whereas $\boldsymbol{F}_{k-1}^{o,cache(i)}$ denotes that, for $i \notin \mathcal{I}$, the $i$-th row of $\boldsymbol{F}_k^o$ is taken from the corresponding token of $\boldsymbol{F}_{k-1}^{o,cache}$. As shown in Tab. 1⑥, this achieves a $1.8\times$ speedup for VAR inference with nearly negligible additional latency (i.e., $\gtrsim$0s).

For *semantic irrelevance*, we adopt the CFG during the *fidelity refinement stage* by setting it to 0, which is equivalent to conditioning only on the null text prompt. Combining RP with RTR then enables plug-and-play acceleration in VAR. Our full algorithm is presented in Algorithm 1 (Sec. B). See Fig. 8 for more details on the vanilla VAR (Fig. 3①), low-rank strategies (Fig. 3②-④ and Tab. 1⑤), and the proposed method (Fig. 4).

# 4. Experiments

## 4.1. Experimental Setup

We build our method **FasterVAR** on the VAR-based text-to-image model Infinity-2B, Infinity-8B (Han et al., 2025), and STAR-1.7B (Ma et al., 2024b), with images generated at a resolution of $1024 \times 1024$. We then evaluate our method on the GenEval (Ghosh et al., 2023) and DPG (Hu et al., 2024) benchmarks, which are two widely adopted benchmarks for assessing semantic alignment and perceptual quality of generated images (Han et al., 2025; Guo et al., 2025; Li et al., 2025a). Additionally, we use Fréchet Inception

Distance (FID) (Heusel et al., 2017), Kernel Inception Distance (KID) (Bińkowski et al., 2018), and Inception Score (IS) (Barratt & Sharma, 2018) metrics on the widely used COCO 2014 and COCO 2017 benchmarks (Lin et al., 2014) to further evaluate perceptual quality. To evaluate the efficiency of the proposed method, we report the latency and the corresponding speedup ratio.

Based on our analysis, we preserve the original inference process in Infinity for the *semantic establishment stage* and *structure establishment stage* (i.e., scales $\{1, 2, 4, 6, 8, 12, 16, 20, 24, 32\}$), while applying acceleration strategies in the *fidelity refinement stage* (i.e., scales $\{40, 48, 64\}$). For the threshold $\alpha$, we set it to $\{0.96, 0, 0\}$ in the *fidelity refinement stage*. Setting $\alpha = 0$ indicates that the corresponding scale step is skipped, and the intermediate result is interpolated to the target resolution as the final output. We set $\alpha$ to 0.96 in the scale $\{64\}$ in STAR (Ma et al., 2024b). We apply our acceleration operation at the block level (e.g., 8 blocks in the Infinity backbone, 30 blocks in the STAR backbone). We use one RTX 3090 GPU (24GB VRAM) to conduct all our experiments, except for Infinity-8B, which is run on an A100 GPU (80 GB VRAM).

## 4.2. Main Results

**Comparison with Baselines.** Tab. 3 presents results on GenEval (Ghosh et al., 2023) and DPG (Hu et al., 2024) benchmarks. As shown in Tab. 3, Infinity (Han et al., 2025) achieves superior performance on both benchmarks within just 13 steps, except for DALL-E 3 on the DPG benchmark, demonstrating advantages over both multi-step diffusion models (Betker et al., 2023; Podell et al., 2024; Chen et al.,

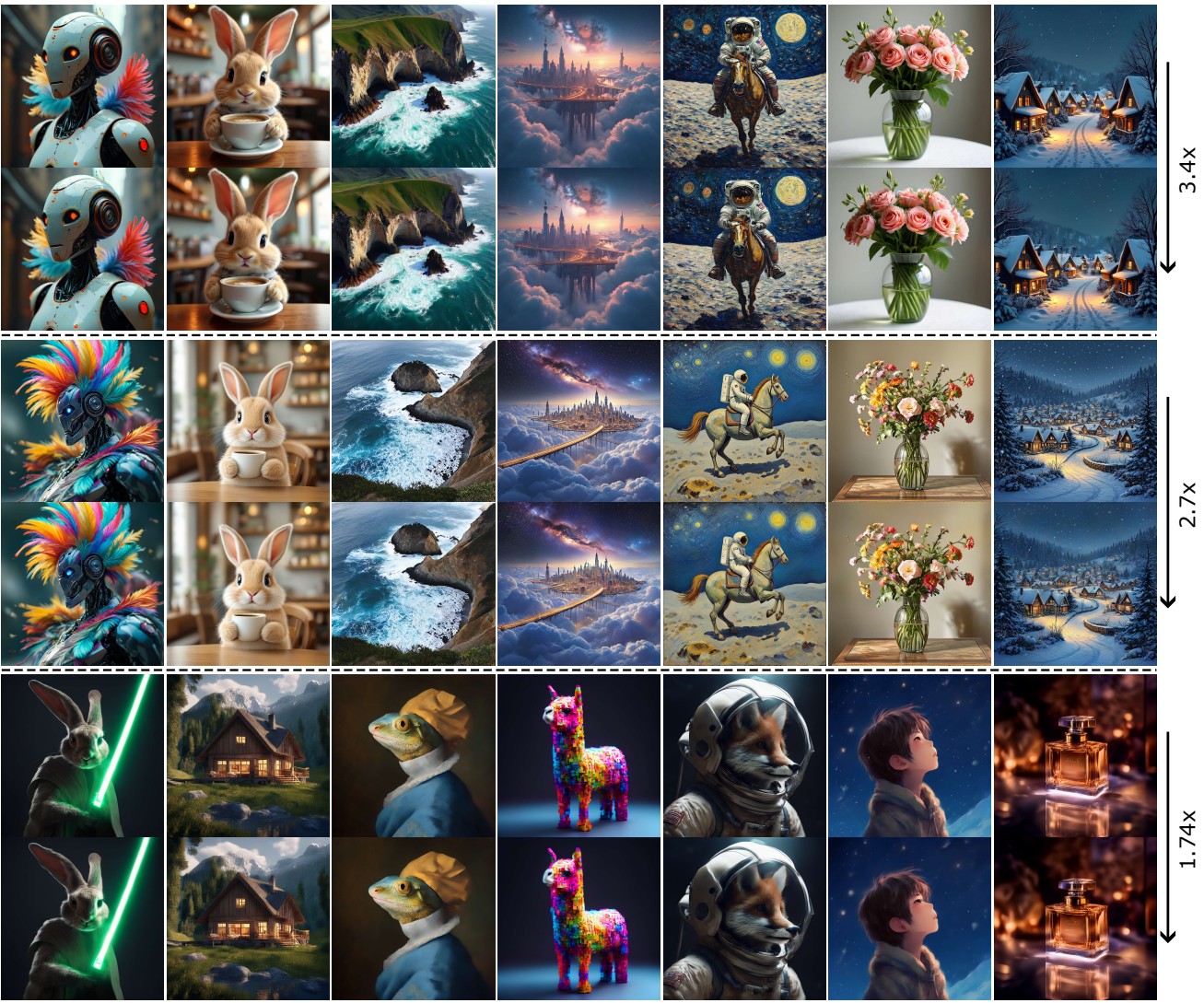

*Figure 5.* Qualitative comparison with the vanilla Infinity-2B, Infinity-8B, and STAR models (1st, 3rd, and 5th rows). Our **FasterVAR** (2nd, 4th, and 6th rows) achieves a $3.4\times$, $2.7\times$, and $1.74\times$ speedup while maintaining performance.

2024) and AR models (Xie et al., 2025; Wang et al., 2024; Tang et al., 2025; Sun et al., 2024). When combined with FastVAR (Guo et al., 2025) and SkipVAR (Li et al., 2025a), the vanilla models achieve $2.75\times$ and $2.62\times$ speedups, respectively. In comparison, integrating **FasterVAR** (Ours) with the Infinity-2B and -8B models yields up to $3.4\times$ and $2.7\times$ speedups, respectively, while incurring only negligible performance degradation, demonstrating superior acceleration over existing methods. In addition, STAR (Ma et al., 2024b) with **FasterVAR** achieves a $1.74\times$ speedup while maintaining performance. As shown in the qualitative evaluation in Fig. 5, ours preserves high visual quality, confirming that the results remain consistent with the vanilla models. Tab. 4 presents results on the COCO2014 and COCO2017 benchmarks (Lin et al., 2014) to further validate perceptual quality. The results in Tab. 4 demonstrate that ours maintains a high speedup ratio while maintaining competitive perfor-

*Table 4.* Quantitative comparison of FID, KID, and IS on COCO2014 and COCO2017.

| Methods | #Speed↑ | COCO2014-30K | | | COCO2017-5K | | |
|---|---|---|---|---|---|---|---|
| | | FID↓ | KID$\times 10^2$ ↓ | IS↑ | FID↓ | KID$\times 10^2$ ↓ | IS↑ |
| Infinity | $1.0\times$ | 26.64 | 1.26 | 42.61 | 35.82 | 1.31 | 37.21 |
| **Ours** | $3.4\times$ | 26.91 | 1.36 | 42.18 | 37.13 | 1.42 | 37.70 |

mance. For instance, ours achieves a $3.4\times$ acceleration with only minor degradations of 1.3 in FID and around 0.5 in IS compared with the vanilla model. We further evaluate our method against baseline approaches using the HPS (Wu et al., 2023) and PickScore (Kirstain et al., 2023) metrics. As shown in Tab. 5, our method maintains aesthetics and human preference.

**Impact of Semantic Irrelevance.** As reported in Tab. 6

*Table 5.* Quantitative comparison of HPSv2.1 and PickScore.

| Methods | #Speed↑ | #Latency↓ | HPSv2.1↑ | PickScore↑ |
|---------|---------|-----------|----------|------------|
| Infinity | 1.0× | 2.2s | 30.48 | 23.12 |
| **Ours** | 3.4× | 0.64s | 30.07 | 23.00 |
| STAR | 1.0× | 2.0s | 29.16 | 21.56 |
| **Ours** | 1.74× | 1.15s | 29.06 | 21.28 |

*Table 6.* Ablation study of incorporating ① CFG=0 and ② low-rank strategy (RP+RTR). ③ FastVAR.

| **Methods** | #Speed↑ | #Latency↓ | **GenEval↑** | **DPG↑** |
|-------------|---------|-----------|--------------|----------|
| Infinity | 1.0× | 2.2s | 0.731 | 83.12 |
| + ① | 1.5× | 1.45s | 0.724 | 82.78 |
| + ① ② (**Ours**) | 3.4× | 0.64s | 0.726 | 82.86 |
| + ① ③ | 3.14× | 0.70s | 0.711 | 82.72 |

(2nd row), applying the semantic irrelevance strategy yields a 1.5× speedup without degrading the quality. Furthermore, combining with random projection (RP) and representative token restoration (RTR) achieves a superior speedup of 3.4× (Tab. 6 (3rd row)). The semantic irrelevance mechanism can be integrated into FastVAR (Tab. 6, 4th row), achieving a 3.14× speedup, though it remains inferior to our method.

**Different Ranks.** Rank is critical for balancing the efficiency and performance of our method. Existing token-reduction text-to-image generation methods achieve faster forward computation by using fewer tokens, but at the cost of degraded performance (Bolya & Hoffman, 2023; Guo et al., 2025; Chen et al., 2025a). Interestingly, unlike these approaches, we find that reducing the rank initially improves performance, reaches a peak, and then degrades as the rank continues to decrease (Fig. 6 (Left)). A qualitative comparison also reveals a similar trend in finer details, such as the illustration of the "mouth" in Fig. 6 (Right). This pattern has also been reported in (Durrant & Kabán, 2013), where performance under random projection peaks at an intermediate projection dimension. Moreover, as the rank decreases, the speedup ratio shows a stable increase (Fig. 6 (the green curve)). Therefore, based on both quantitative and qualitative results, we select a 17.6% rank (i.e., $\alpha$=0.96) for our acceleration method **FasterVAR**.

**Additional Results.** The Infinity (Han et al., 2025) model originally supports image generation with varying aspect ratios and our method **FasterVAR** supports this property. As shown in Fig. 7, when combined with **FasterVAR**, it can still facilitate efficient image generation, indicating that our proposed **FasterVAR** can be easily extended to generate images with diverse aspect ratios.

# 5. Conclusion

In this work, we address the computational inefficiency of visual autoregressive (VAR) models via a systematic anal-

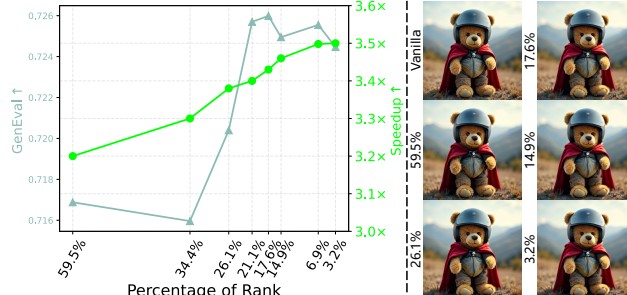

*Figure 6.* Visualization of the quantitative and qualitative results for different ranks.

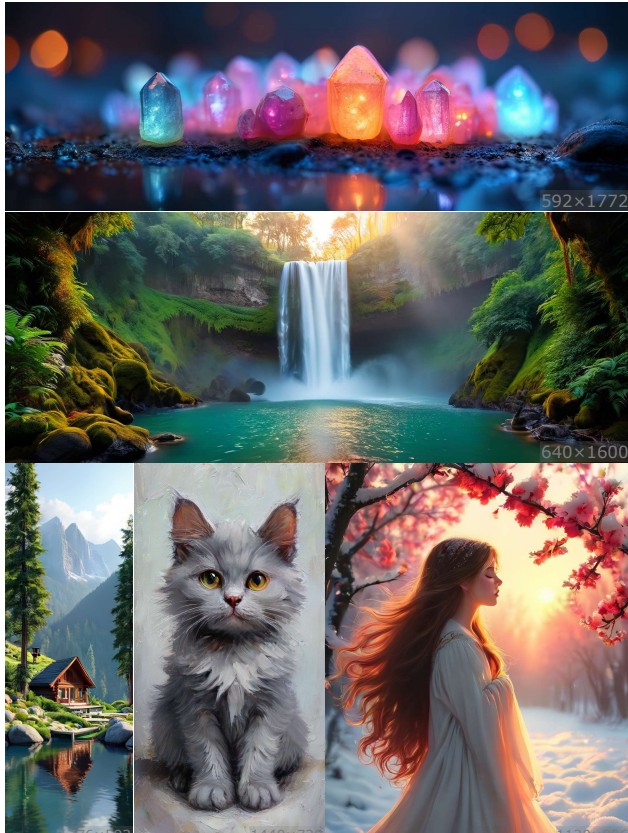

*Figure 7.* Qualitative results of **FasterVAR** with diverse aspect ratios.

ysis of their inference process. We identify three distinct stages—semantic establishment, structure establishment, and fidelity refinement—showing that early steps secure core content while later steps only refine details. Leveraging this insight, we propose **FasterVAR**, a plug-and-play, training-free acceleration method that exploits semantic irrelevance (bypassing text conditioning) and low-rank features (reducing feature space) in the fidelity refinement stage. Benchmark experiments validate our method, achieving a 3.4× speedup over baselines with negligible performance drop. This work advances efficient VAR inference, offering a practical solution to balance speed and quality.

## Acknowledgements

This work was supported by the NSFC under Grant Nos. 62361166670 and U24A20330. This work was also supported by the Google Gemini Academic Program Award. Kai Wang acknowledges the funding No.R002026G0116 and No.R002026B0121 from Guangdong and Hong Kong Universities 1+1+1 Joint Research Collaboration Scheme, and the start-up grant B01040000108 from CityU-DG. Computation is supported by the Supercomputing Center of Nankai University (NKSC).

## Impact Statement

This paper presents work whose goal is to advance the field of Machine Learning. There are many potential societal consequences of our work, none which we feel must be specifically highlighted here.

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

# A. Implementation details

## A.1. Statistical Analysis of the Rank

In order to determine the rank $r$ corresponding to a given $\alpha$ satisfying Eq. (4), it is necessary to perform an SVD decomposition of the original feature $\widetilde{F}_{k-1}$ to obtain the energy ratio $\eta_r$, where the SVD decomposition is time-consuming. In Sec. 3.3, to address the additional time required to determine $r$ for a given $\alpha$, we adopt an off-the-shelf value of $r$ corresponding to $\alpha$, thereby avoiding both SVD decomposition and the use of Eq. (4). Specifically, we adopt 553 prompts from the GenEval benchmark (Ghosh et al., 2023). For each prompt, four images are randomly generated. During the generation process, we perform SVD decomposition on the features of each input block to obtain $\eta_r$, and then compute the corresponding rank $r$ for a given $\alpha$ using Eq.4. The collected $r$ values across all prompts are then aggregated to determine a representative rank for each $\alpha$. In this way, once $\alpha$ is specified, the corresponding $r$ can be directly assigned. For example, as shown in Tab. 7, when $\alpha = 0.96$, the standard deviation of $r$ across different blocks is an order of magnitude smaller than its mean, suggesting that for a given $\alpha$, the features exhibit stable low-rank characteristics across diverse text prompts.

To further demonstrate that the statistical rank $r$ for a given $\alpha$ generalizes across different text prompts, we additionally perform the analysis on the COCO2014 and COCO2017 datasets, which offer more diverse and broader descriptions (Lin et al., 2014). Specifically, we randomly sample 1K text prompts from COCO2014 and generate one image for each prompt. Similarly, 1K text prompts are randomly selected from COCO2017, with one image generated per prompt. The statistical results are shown in Tabs. 8 and 9. The deviation from the corresponding block and scale values reported in Tab. 7 is negligible, confirming that the statistical rank $r$ for a given $\alpha$ generalizes robustly across diverse text prompts.

Note that while collecting the statistical results on the benchmark is computationally expensive, it is an offline, one-time process. During inference, the pre-determined rank $r$ can be directly applied base on a given $\alpha$ without incurring additional overhead.

*Table 7.* Detailed information about the ranks of the block features in Infinity (Han et al., 2025).

| block name \ $k$-th scale | 40 | 48 | 64 |
|---|---|---|---|
| block_chunks.0 | 0.016±0.0004 | 0.013±0.0003 | 0.008±0.0007 |
| block_chunks.1 | 0.136±0.0096 | 0.117±0.0078 | 0.054±0.0064 |
| block_chunks.2 | 0.210±0.0187 | 0.189±0.0161 | 0.083±0.0176 |
| block_chunks.3 | 0.250±0.0212 | 0.222±0.0171 | 0.066±0.0164 |
| block_chunks.4 | 0.256±0.0236 | 0.234±0.0180 | 0.056±0.0146 |
| block_chunks.5 | 0.198±0.0170 | 0.183±0.0124 | 0.029±0.0084 |
| block_chunks.6 | 0.157±0.0074 | 0.134±0.0053 | 0.038±0.0067 |
| block_chunks.7 | 0.191±0.0068 | 0.163±0.0057 | 0.055±0.0142 |

*Table 8.* Detailed information about the ranks of the block features in Infinity (Han et al., 2025) on the COCO2014 dataset.

| block name \ $k$-th scale | 40 | 48 | 64 |
|---|---|---|---|
| block_chunks.0 | 0.016±0.0004 | 0.013±0.0003 | 0.006±0.0003 |
| block_chunks.1 | 0.142±0.0087 | 0.121±0.0068 | 0.054±0.0057 |
| block_chunks.2 | 0.242±0.0168 | 0.213±0.0145 | 0.097±0.0168 |
| block_chunks.3 | 0.299±0.0188 | 0.256±0.0155 | 0.051±0.0185 |
| block_chunks.4 | 0.317±0.0192 | 0.278±0.0155 | 0.062±0.0177 |
| block_chunks.5 | 0.239±0.0132 | 0.212±0.0099 | 0.040±0.0092 |
| block_chunks.6 | 0.186±0.0059 | 0.157±0.0043 | 0.048±0.0033 |
| block_chunks.7 | 0.232±0.0062 | 0.198±0.0056 | 0.069±0.0060 |

## A.2. Construction of $\mathrm{U}_r^o$

As shown in Fig. 3③ in Sec. 3.3, we construct the $r$-dimensional feature $\widetilde{F}_r = \widetilde{\Sigma}_r \widetilde{V}_r^T$. $\widetilde{F}_r$ serves as an $r$-dimensional representation of the row space of $\widetilde{F}_{k-1}$, conditioned on $\widetilde{U}_r$ to provide the most closely rank-$r$ approximation (Eckart

*Table 9.* Detailed information about the ranks of the block features in Infinity (Han et al., 2025) on the COCO2017 dataset.

| block name \ $k$-th scale | 40 | 48 | 64 |
|---|---|---|---|
| block_chunks.0 | 0.017±0.0004 | 0.013±0.0003 | 0.007±0.0002 |
| block_chunks.1 | 0.142±0.0090 | 0.121±0.0068 | 0.054±0.0057 |
| block_chunks.2 | 0.242±0.0174 | 0.213±0.0145 | 0.101±0.0167 |
| block_chunks.3 | 0.299±0.0190 | 0.256±0.0154 | 0.050±0.0181 |
| block_chunks.4 | 0.318±0.0193 | 0.279±0.0150 | 0.062±0.0167 |
| block_chunks.5 | 0.239±0.0138 | 0.213±0.0099 | 0.038±0.0088 |
| block_chunks.6 | 0.187±0.0061 | 0.157±0.0044 | 0.047±0.0032 |
| block_chunks.7 | 0.232±0.0064 | 0.198±0.0059 | 0.067±0.0060 |

& Young, 1936). However, after the model forward, $\widetilde{F}_r$ produces only $F_r^o$ (Fig. 3③), while $\mathbf{U}_r^o$ is unavailable, thereby hindering the restoration of the original $M$-dimensional feature $F_k^o$. According to Eq. (2), the autoregressive likelihood models the token at the $k$-th scale step as conditioned not only on itself but also on all previous steps $\{1, 2, \ldots, k\text{-}1\}$. Inspired by this mechanism, FastVAR (Guo et al., 2025) caches outputs from the $(k\text{-}1)$-th scale step to restore the tokens for the $k$-th scale. Similarly, we employ $\mathbf{U}_r^{o,cache}$ from the cached counterpart feature $F_{k-1}^o$ to compensate for $\widetilde{\mathbf{U}}_r$, as follows $\mathbf{U}_r^o \approx (\widetilde{\mathbf{U}}_r + \mathbf{U}_r^{o,cache})$. In detail, the cached feature $F_{k-1}^o$ is upsampled to match the dimensions of $F_k^o$ as $F_{k-1}^{o,cache}$ (See Eq. (7)), followed by SVD to obtain $\mathbf{U}_r^{o,cache}$.

## A.3. Construction of $\mathbf{W}_r^o$

As shown in Fig. 4 (Fidelity refinement stage), after the model forward, $\widehat{F}_r$ produces only $F_r^o$, while $\mathbf{W}_r^o$ is unavailable, thereby hindering the restoration of the original $M$-dimensional feature $F_k^o = \mathbf{W}_r^o F_r^o$. Similarly to the naive SVD-based strategy (Tab. 1③ and Fig. 3③) in Sec. A.2, we leverage the $\mathbf{W}_r^{o,cache}$ from the cached counterpart feature $F_{k-1}^o$ to compensate for $\widehat{\mathbf{W}}_r$, as follows $\mathbf{W}_r^o \approx (\widehat{\mathbf{W}}_r + \mathbf{W}_r^{o,cache})$. In detail, the cached token map $F_{k-1}^o$ is upsampled to match the dimensions of $F_k^o$ as $F_{k-1}^{o,cache}$ (See Eq. (7)), followed by solving LLS problem to obtain $\mathbf{W}_r^{o,cache}$.

## A.4. Details of Different Strategies

As shown in Fig. 8, we provide additional details on the vanilla VAR framework corresponding to Fig. 3① in Fig. 8①; the SVD-based low-rank strategies corresponding to Fig. 3②–④ and Tab. 1②–④ in Fig. 8②–④; the LLS-based restoration method corresponding to Tab. 1⑤ in Fig. 8⑤; and the proposed **FasterVAR** corresponding to Fig. 4 in Fig. 8⑥.

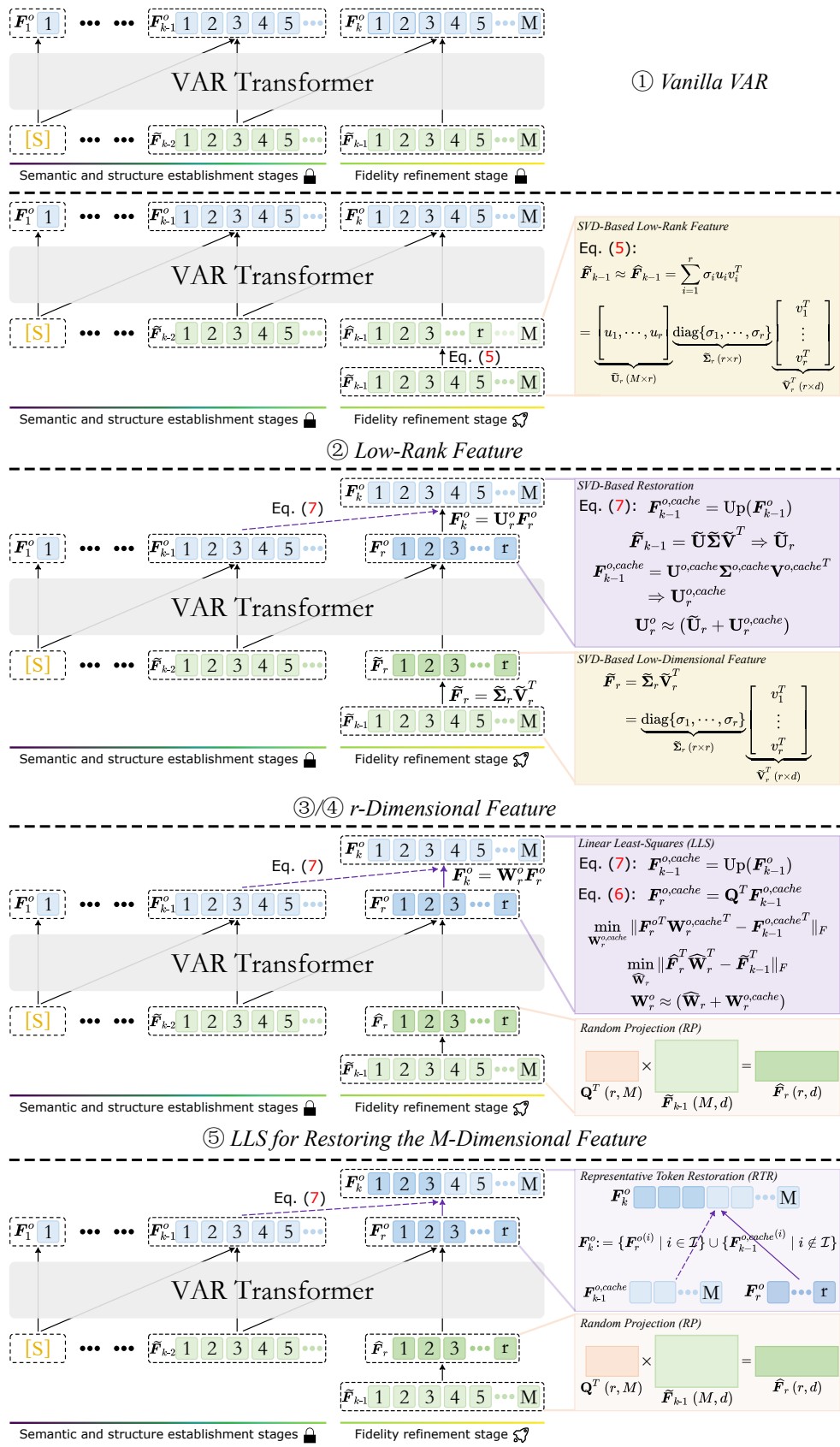

*Figure 8.* Overview of vanilla VAR (①), low-rank strategies (②-⑤), and the proposed **FasterVAR** (⑥).

# B. Algorithm detail of FasterVAR

---

**Algorithm 1** : **FasterVAR**

---

**Input** : Scale steps $\{1, 2, \cdots, K\}$. Scale steps set of the *fidelity refinement stage* consists of $m$ steps $\{K\text{-}m\text{+}1, \cdots, K\}$. Steps set of the *semantic and structure establishment stages* $\{1, \cdots, K - m\}$. The VAR model $\phi$, and the image decoder $\mathcal{D}$. The quantizer $\mathcal{Q}$, which typically includes a codebook $Z \in \mathbb{R}^{V \times d}$ containing $V$ vectors.

**Output** : The final generated images $\mathbf{I}$

---

$\boldsymbol{F}_0 = 0$ ;
$\widetilde{\boldsymbol{F}}_0 = \langle \text{SOS} \rangle \in \mathbb{R}^{1 \times 1 \times d}$ ;                              `// ⟨SOS⟩ is the start token (Han et al., 2025)`
`// the semantic and structure establishment stages`
**for** $k = 1, \cdots, K\text{-}m$ **do**
  $\boldsymbol{F}_k^o = \phi(\widetilde{\boldsymbol{F}}_{k-1})$ ;                                                            `// Fig. 3①`
  $\boldsymbol{R}_k = \mathcal{Q}(\boldsymbol{F}_k^o)$ ;
  $\boldsymbol{F}_k = \boldsymbol{F}_{k-1} + \text{Up}(\boldsymbol{R}_k, (h_K, w_K))$ ;                                        `// Eq. (1)`
  $\widetilde{\boldsymbol{F}}_k = \text{Down}(\boldsymbol{F}_k, (h_k, w_k))$ ;                                               `// Eq. (3)`
**end**
`// the fidelity refinement stage (CFG=0)`
**for** $k = K\text{-}m\text{+}1, \cdots, K$ **do**
  $\widehat{\boldsymbol{F}}_r = \mathbf{Q}^T \widetilde{\boldsymbol{F}}_{k-1}$ ;                     `// Random Projection (RP) for Low-Rank Feature` $\widehat{\boldsymbol{F}}_r$
  $\boldsymbol{F}_r^o = \phi(\widehat{\boldsymbol{F}}_r)$ ;                                                                 `// Fig. 4`
  `// Representative Token Restoration (RTR)`
  $\boldsymbol{F}_{k-1}^{o,cache} = \text{Up}(\boldsymbol{F}_{k-1}^o)$ ;                                                    `// Eq. (7)`
  The chosen indices $\mathcal{I}$, based on $\widetilde{\boldsymbol{F}}_{k-1}$ ;                                  `// based on (Frieze et al., 2004)`
  $\boldsymbol{F}_k^o := \{\boldsymbol{F}_r^{o(i)} \mid i \in \mathcal{I}\} \cup \{\boldsymbol{F}_{k-1}^{o,cache(i)} \mid i \notin \mathcal{I}\}$ ;    `// Eq. (8)`
  `// Vanilla VAR`
  $\boldsymbol{R}_k = \mathcal{Q}(\boldsymbol{F}_k^o)$
  $\boldsymbol{F}_k = \boldsymbol{F}_{k-1} + \text{Up}(\boldsymbol{R}_k, (h_K, w_K))$ ;                                        `// Eq. (1)`
  $\widetilde{\boldsymbol{F}}_k = \text{Down}(\boldsymbol{F}_k, (h_k, w_k))$ ;                                               `// Eq. (3)`
**end**
$\mathbf{I} = \mathcal{D}(\boldsymbol{F}_K)$
**Return** The final generated image $\mathbf{I}$

---

# C. Ablation analysis

### C.1. Robustness to Random Projection (RP)

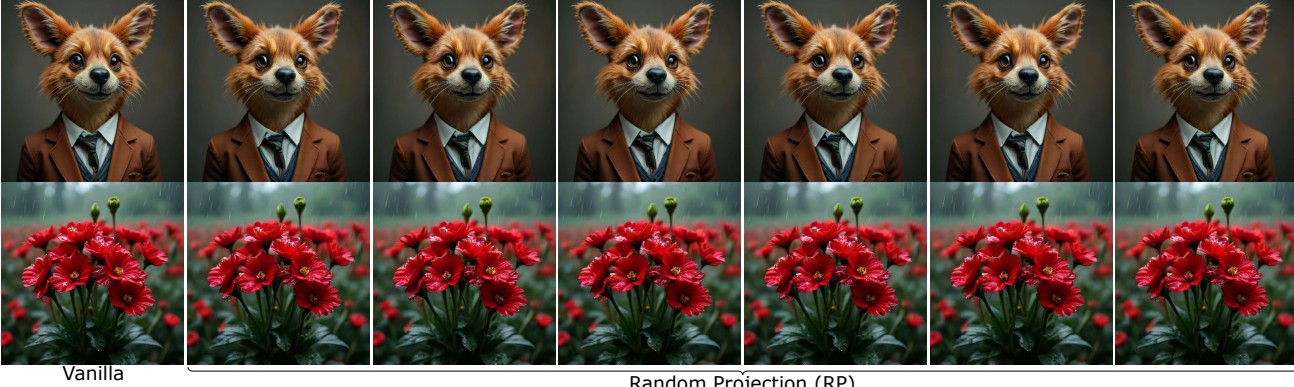

Vanilla                        Random Projection (RP)

*Figure 9.* Two examples of our **FasterVAR**, which achieves $3.4\times$ speedup while consistently producing images of comparable quality to Vanilla (i.e., Infinity (Han et al., 2025)) across multiple generations with random projection (RP).

As shown in Fig. 9, we demonstrate the robustness of our method to random projection (RP) when obtaining the $r$-dimensional feature. **FasterVAR** achieves significant sampling speedup (3.4×) while maintaining image quality.

## C.2. Frequency-domain Analysis

To evaluate the low- and high-frequency components of the generated images compared with those of the vanilla model, we conduct experiments in the frequency domain (Fig. 10). Specifically, we use 553 prompts from the GenEval benchmark, with both Infinity and **FasterVAR** generating one random image per prompt. As shown in Fig. 10 (Left), we observe that the low- and high-frequency components exhibit almost no loss in the frequency domain, suggesting strong frequency-domain consistency between methods. This result ensures that the generated images closely resemble those of both Infinity and **FasterVAR** (with the proposed index sampling strategy), while maintaining the desired fidelity. The qualitative comparison further supports this consistency, showing that the high- and low-frequency components exhibit almost no noticeable differences across methods (Fig. 10 (Right)).

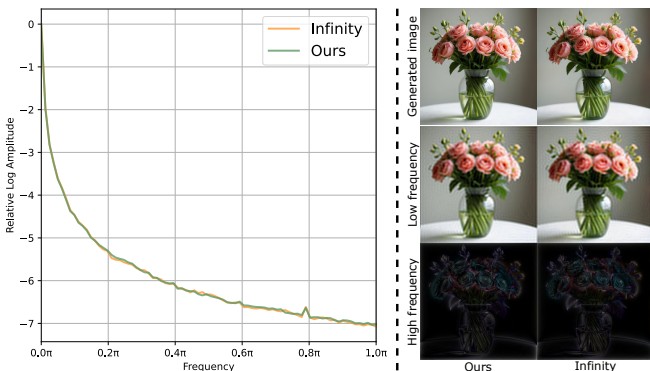

*Figure 10.* The low- and high-frequency components of the generated images from Ours closely resemble those from the vanilla model (Infinity).

## C.3. Semantic Irrelevance in Complex Prompts

Fig. 11 shows that **FasterVAR** maintains both generation quality and text–image alignment even for complex prompts. As shown in Fig. 11 (the last column (Prompt⑤)), it can still preserve semantic consistency even when using rare text prompts.

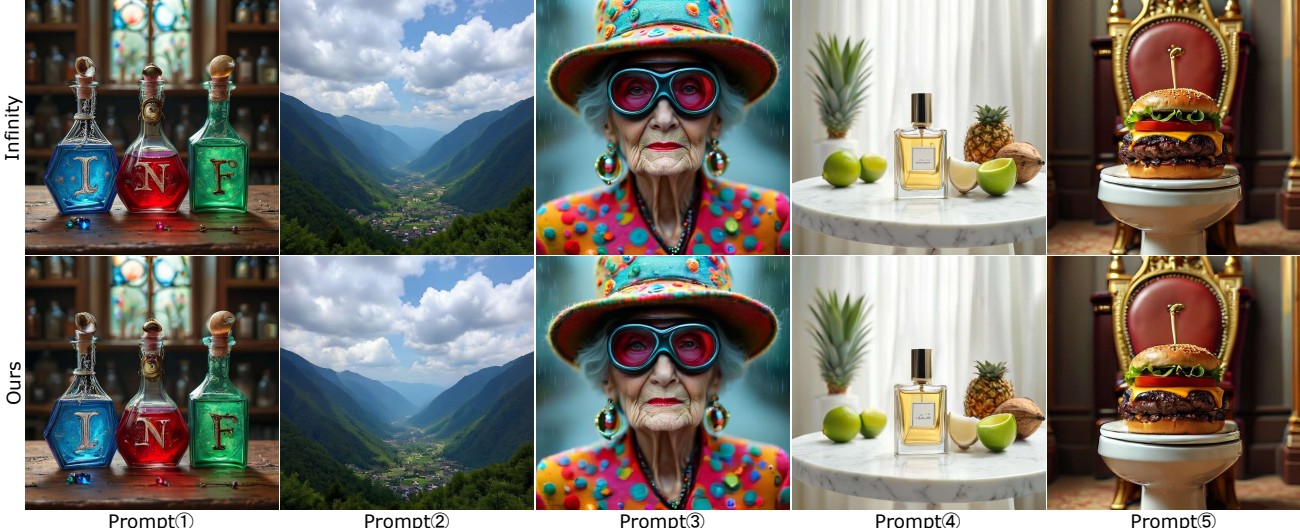

*Figure 11.* **FasterVAR** maintains both generation quality and text–image alignment even for complex prompts.

**Prompt①:** Create a mesmerizing image of three intricately designed potions displayed on an ornate, antique wooden table within a charming old apothecary. The first potion is a captivating cobalt blue, housed in a stunning pentagon-shaped glass bottle that sparkles with its many facets; its label, meticulously crafted with delicate silver filigree and botanical illustrations of ethereal flowers, prominently features the letters "I" in an ornate, swirling script, while a silver ribbon interwoven with tiny sapphire beads wraps around the neck, adorned with a charm in the shape of a crescent moon. The second potion is a rich crimson red, contained in a flat, oval-shaped glass bottle adorned with

intricate engravings of mystical symbols, including runes and ancient scripts; its label displays the letters "N" in embossed gold leaf, framed by elaborate floral designs, and is topped with a cork stopper embellished with a miniature brass key and tiny ruby gemstones. The third potion is a vivid emerald green, held in a sleek square glass bottle featuring enchanting etchings of mythical creatures like dragons and phoenixes; its scroll-like label, crafted from aged parchment, prominently features the letter "F" intertwined with ancient alchemical symbols and delicate vine patterns. All three bottles are approximately the same height, creating a harmonious display against a backdrop filled with shelves overflowing with dried herbs, colorful glass jars, and ancient scrolls, all illuminated by soft, warm light filtering through a stained-glass window, enhancing the magical atmosphere of the apothecary.

**Prompt②:** The image presents a picturesque mountainous landscape under a cloudy sky. The mountains, blanketed in lush greenery, rise majestically, their slopes dotted with clusters of trees and shrubs. The sky above is a canvas of blue, adorned with fluffy white clouds that add a sense of tranquility to the scene. In the foreground, a valley unfolds, nestled between the towering mountains. It appears to be a rural area, with a few buildings and structures visible, suggesting the presence of a small settlement. The buildings are scattered, blending harmoniously with the natural surroundings. The image is captured from a high vantage point, providing a sweeping view of the valley and the mountains.

**Prompt③:** A bullet time photography of a beautiful 90 year old woman, frontal portrait, colorful futuristic felt dress covered with colorful buttons, futuristic hat, Gucci style earrings, very bright red lipstick, the woman is wearing a pair of goggles, colored rain.

**Prompt④:** Product photography, a perfume placed on a white marble table with pineapple, coconut, limenext to it as decoration, white curtains, full of intricate details, realistic, minimalist, layered gestures in a bright and concise atmosphere, minimalist style.

**Prompt⑤:** A cheeseburger with juicy beef patties and melted cheese sits on top of a toilet that looks like a throne and stands in the middle of the royal chamber.

## D. Additional analysis

### D.1. User Study.

We conducted a user study, as shown in Fig. 12, and asked subjects to select results. We apply pairwise comparisons (forced choice) with 69 users (42 pairs of images). The results demonstrate that our method performs equally well as the vanilla models in terms of human preference.

### D.2. Quality-Latency Comparison

To further evaluate the generation quality of **FasterVAR**, we conduct experiments to investigate the relationship between FID and inference speed on the COCO 2017 dataset. As shown in Fig. 13, the performance of **FasterVAR** initially improves with increasing speed, reaches its optimal point at a speedup of $3.4\times$, and then degrades as the speed continues to increase, following a trend similar to the GenEval metrics observed in Fig. 6 (Left). In contrast, FastVAR exhibits a consistent degradation in FID as the inference speed increases—although a slight improvement is observed at $3.06\times$—its FID remains above 9.7 across all settings. These results indicate that our method achieves a better balance between quality and speed compared to FastVAR.

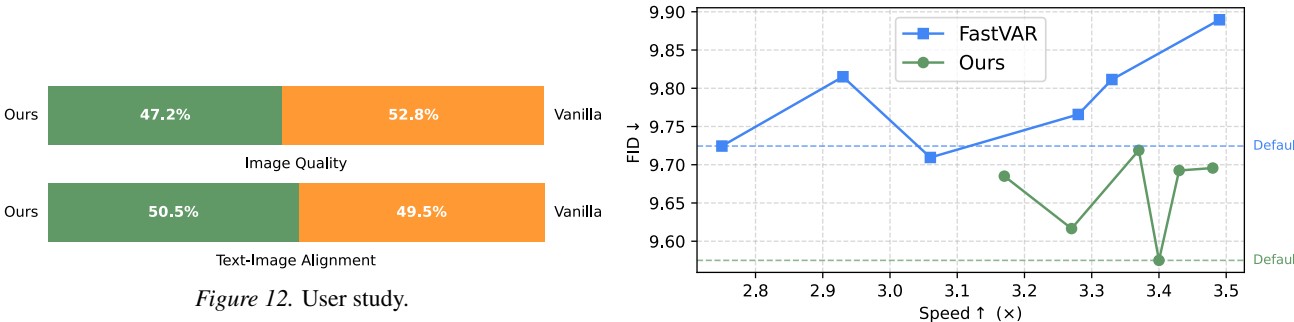

*Figure 12.* User study.

*Figure 13.* Comparison of quality (FID) and inference speed between FastVAR and Ours.

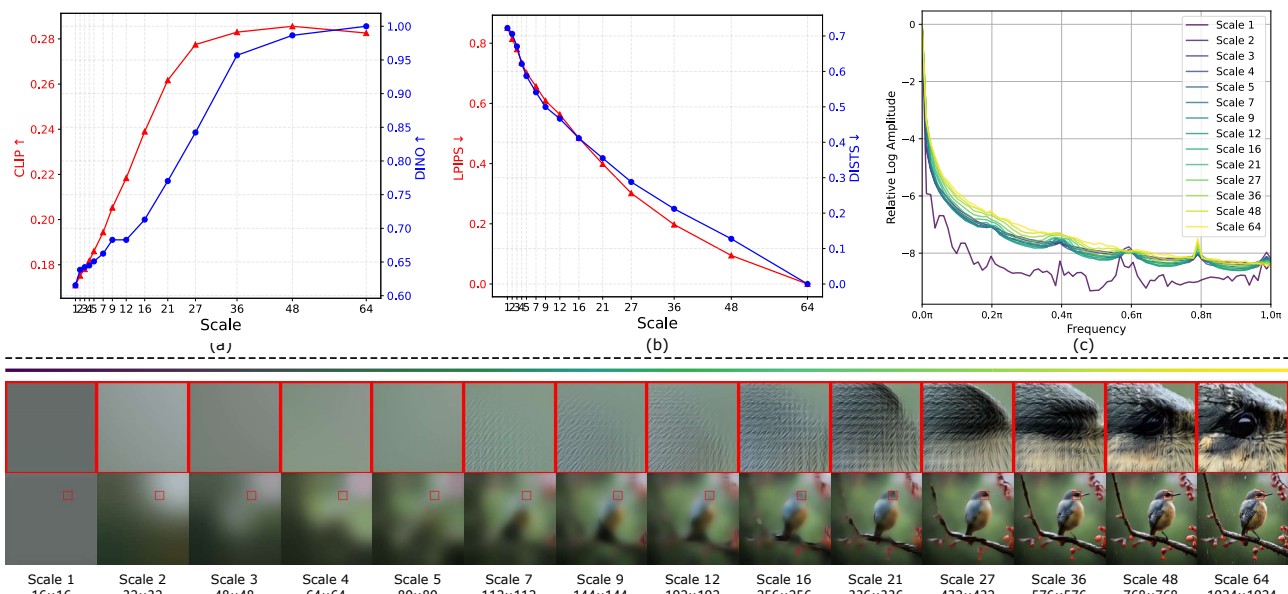

*Figure 14.* (a) Visualization of semantic evolution across all scale steps (i.e., CLIP and DINO). (b) Visualization of structure evolution on all scale steps (i.e., LPIPS and DISTS). (c) Variations of the next scale step in the frequency domain. (Bottom) Visualization of samples across all scale steps in STAR (Ma et al., 2024b).

## D.3. FasterVAR for Other Text-to-Image VAR Models: STAR (Ma et al., 2024b)

We evaluate semantic (CLIP and DINO) and structural consistency (LPIPS and DISTS) across all scale steps in the next-scale prediction model, STAR (Ma et al., 2024b). As shown in Fig. 14 (a-b), local semantic (DINO) and structural consistency (LPIPS and DISTS) are gradually established from the initial to the final scale, except that global semantics (CLIP) converge at the later scales (e.g., scale 48). Fig. 14 (Bottom) also illustrates this tendency. Moreover, both the low-frequency and high-frequency components noticeable variations across all scale steps (Fig. 14c).

We evaluate the semantic irrelevance in the next-scale prediction model STAR (Ma et al., 2024b) by setting the CFG scale to 0 starting from step $k$. As shown in Fig. 15, the CLIP and GenEval scores exhibit convergence at the last two scales. We also evaluate the low-rank property in Tab. 12.

Based on the aforementioned analysis, STAR (Ma et al., 2024b) shows the low-rank property and semantic irrelevance.

## E. Text Prompts

We list the text prompts used for image generation in this paper below.

Fig. 1 (Bottom): "Portrait of an old sea captain, male, detailed face, fantasy, highly detailed, cinematic, art painting by greg rutkowski ".

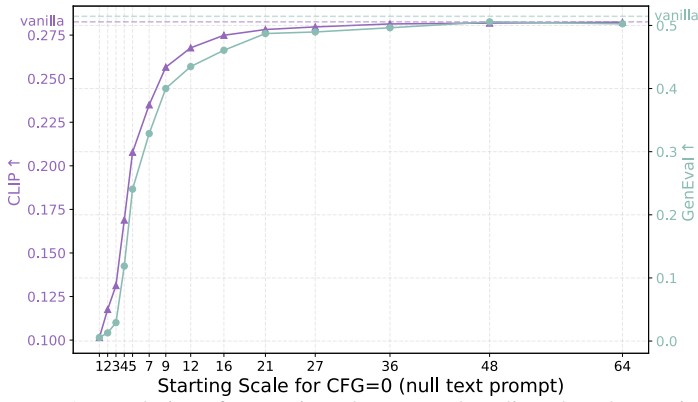

*Figure 15.* Evolution of semantic and perceptual quality when the starting scale steps of CFG is set to 0 for STAR (Ma et al., 2024b)

*Table 12.* Performace with the low-rank feature by varing $\alpha$ to evaluate the low-rank property in STAR (Ma et al., 2024b)

| Methods | GenEval↑ |
|---|---|
| Vanilla | 0.514 |
| $\alpha = 0.999$ (59.5% rank) | 0.510 |
| $\alpha = 0.99$ (34.4% rank) | 0.505 |
| $\alpha = 0.98$ (26.1% rank) | 0.507 |
| $\alpha = 0.97$ (21.1% rank) | 0.505 |
| $\alpha = 0.96$ (17.6% rank) | 0.495 |
| $\alpha = 0.95$ (14.9% rank) | 0.493 |

Fig. 2 (Right): "A smiling girl with glasses wears gemstone earrings and a red scarf in a snowstorm".

Fig. 5: "A cinematic shot of robot with colorful feathers", "A photo of a cute rabbit holding a cup of coffee in a cafe", "Drone view of waves crashing against the rugged cliffs in Big Sur", "A city in the clouds, connected by bridges of light with a galaxy backdrop", "An astronaut riding a horse on the moon, oil painting by Van Gogh", "A vase sitting on top of a table with flowers in it", "A winter village at night, warm light shining from windows, snowflakes gently falling", "A jackrabbit with a green lightsaber, intense expression, jedi", "A beautiful cabin in Attersee, Austria, 3d animation style", "An oil painting of an anthropomorphic bearded dragon in the style of Girl with a Pearl Earring by Johannes Vermeer", "A alpaca made of colorful building blocks, cyberpunk", "A hyper-realistic photo of a fox astronaut, perfect face, artstation", "Cute boy, hair looking up to the stars, snow, beautiful lightening, painting style style abe Toshiyuki", and "Retangular perfume on a camarin, white lights, bright, well lit enviroment, bright lights".

Fig. 6 (Right): "A teddy bear wearing a motorcycle helmet and cape, with a scenic landscape in the background".

Fig. 7: "A beautiful cabin in Attersee, Austria, 3d animation style", "Glowing crystals of different colors on wet surface, macro, bokeh background, mystical, cinematic lighting", " majestic waterfall cascading down a moss-covered cliff in a lush, misty rainforest at golden hour, sunlight filtering through the canopy and creating soft rays and ethereal glows. Crystal-clear water plunges into a turquoise pool below, surrounded by smooth stones and vibrant ferns", "Cute grey cat, digital oil painting by Monet", and "Amidst red plum blossoms and white snow, a fair-haired maiden stands serene. The setting sun gilds her silhouette as a gentle breeze stirs her flowing hair and drifting petals—a moment stolen from a dream. With closed eyes and bowed head, she breathes in the frost-kissed fragrance, her lashes trembling with the warmth of the fading light. Winter's hush, nature's blush; she becomes part of the painting".

Fig. 9: "professional portrait photo of an anthropomorphic" and "a bunch of red flowers that are in the rain".

Fig. 10: "A vase sitting on top of a table with flowers in it".

Fig. 11: See Sec. C.3.

Fig. 14 (Bottom): "A bird perched on a tree branch in the rain".

