# OpenReview forum: "FasterVAR: Plug-and-Play Acceleration for Visual Autoregressive Models"
_ICML.cc/2026/Conference — ICML 2026 regular_

### Official Review · Reviewer_7web · 2026-02-26

**Soundness:** 3
**Presentation:** 4
**Significance:** 3
**Originality:** 3
**Overall Recommendation:** 4
**Confidence:** 5

**Summary:**

This paper targets the inference inefficiency of Visual Autoregressive (VAR) models, which suffer from disproportionately high computational cost at large-scale generation steps. The authors conduct a systematic empirical analysis of the VAR inference process and identify three functionally distinct stages: a semantic establishment stage, a structure establishment stage, and a fidelity refinement stage. Their central finding is that only the final stage where large-scale steps add fine-grained texture rather than meaningful semantic or structural content is amenable to approximation without perceptual degradation. Building on this, they propose FasterVAR, a training-free, plug-and-play acceleration framework that exploits two properties specific to the fidelity refinement stage. First, since text conditioning has negligible influence at this stage, classifier-free guidance can be disabled and replaced with a null prompt, eliminating redundant conditional computations. Second, since the intermediate feature tensors in this stage exhibit a strong low-rank structure, the authors replace them with compact random projections and recover the full-resolution output by filling non-representative tokens with cached features from the previous scale step, avoiding costly SVD at inference time. Applied to popular VAR models, FasterVAR can achieve up to 3.4$\times$ speedup with only marginal performance loss, highlighting stage-aware design as a powerful principle for efficient visual autoregressive image generation.

**Compliance With Llm Reviewing Policy:**

Affirmed.

**Key Questions For Authors:**

* **Q1**: Please refer to the first bullet of the Weaknesses.
* **Q2**: Please refer to the second bullet of the Weaknesses.
* **Q3**: The reported speedup varies substantially across models: 3.4× on Infinity-2B but only 1.74× on STAR-1.7B. The paper applies the fidelity refinement stage acceleration to scales {40, 48, 64} for Infinity but only scale {64} for STAR, without explaining this design choice. Was this boundary determined by the same empirical analysis (e.g., CLIP/DINO/LPIPS curves) applied to STAR, or was it chosen differently? If the refinement stage in STAR genuinely starts later, what accounts for this difference? Clarifying this would help assess whether the three-stage decomposition generalizes across VAR architectures or requires model-specific tuning.

**Limitations:**

Yes.

**Strengths And Weaknesses:**

**Strengths:**
* The paper addresses a practically important topic of improving the inference efficiency of VAR models. The decomposition of VAR inference into three functionally distinct stages is a practically valuable contribution that offers principled guidance for acceleration and may inform future research beyond this work.
* The two acceleration strategies of disabling CFG and applying low-rank random projection with representative token restoration in the fidelity refinement stage are intuitively motivated and grounded in careful empirical analysis.
* The paper is clearly written and well structured.
* The experiments are comprehensive, spanning multiple VAR model families and including thorough ablations on rank threshold, stage boundary, and individual components. The reported 3.4× speedup with negligible performance loss is compelling.

**Weaknesses:**
* The core observation that semantics and structure are already well-established before the fidelity refinement stage (as shown in Figure 1(a)) also implies that the chosen evaluation benchmarks — GenEval and DPG — are largely insensitive to what happens in this final stage. These benchmarks primarily measure semantic alignment and high-level perceptual quality, so it is unsurprising that approximating or accelerating the fidelity refinement stage incurs negligible metric degradation. This raises a more fundamental question: if the fidelity refinement stage contributes little to semantics and structure, a simpler and potentially competitive baseline would be to skip it entirely and instead apply a lightweight super-resolution module to upsample the lower-resolution output from the structure establishment stage. The paper does not compare against such a baseline, making it difficult to assess whether the proposed acceleration strategies offer any advantage over simply generating at a coarser scale and upsampling.
* The computational advantage of skipping or approximating the fidelity refinement stage may be partially attributable to the fact that, at high resolution (1024×1024), the earlier stages already account for a large number of tokens, making the relative contribution of the refinement stage to the generation performance naturally smaller. It remains unclear whether the same negligible performance drop would hold at lower resolutions (e.g., 256×256), where the fidelity refinement stage represents a more dominant factor and may carry more perceptual significance. The absence of any low-resolution experiments leaves this an open question and weakens the generality of the claimed approach.

---

> ### Author Rebuttal · Authors · 2026-03-31
>
> We sincerely thank the reviewer 7web for the thoughtful and constructive comments.
>
> We use **W** to abbreviate Weaknesses, and **Q** to represent Questions.
>
> **W1&Q1: Comparison between the proposed method and coarse generation with upsampling.**
>
> In addition to using GenEval and DPG for evaluation, we further report FID, KID, and IS metrics in Table 4. The metrics only show a slight drop, the results demonstrate that our method maintains competitive performance in terms of fidelity.
> To further evaluate the role of the fidelity refinement stage, we skip this stage and directly upsample the output from the structure establishment stage. As shown in the table below, the GenEval and DPG metrics remain aligned with Infinity, whereas the FID, KID, and IS metrics show inferior performance,  and the processing time increases significantly. This performance drop demonstrates that the fidelity refinement stage plays an essential role and cannot be substituted by super-resolution upsampling alone; dedicated processing at this stage is necessary.
>
> |Methods|#Latency↓|GenEval↑|DPG↑|FID↓|KID×10²↓|IS↑|
> |-|-|-|-|-|-|-|
> |Infinity|2.2s|0.73|83.12|26.64|1.26|42.61|
> |Infinity+Real-ESRGAN|1.21s|0.71|82.82|27.80|1.44|40.43|
> |Infinity+StableSR|4.4s|0.72|82.32|28.25|1.48|39.20|
> |Infinity+SwinIR|2.3s|0.71|83.09|27.47|1.40|41.34|
> |Ours|0.64s|0.72|82.86|26.91|1.36|42.18|
>
> **W2&Q2(1): Is the contribution of the refinement stage naturally smaller?**
>
> Based on our analysis, the fidelity refinement stage mainly refine details and plays an important contribution. If this stage is skipped and the output from the structure establishment stage is directly upsampled using super-resolution methods, the generated fidelity does not align with that of Infinity. Furthermore, when we skip the fidelity refinement stage, we observe a degradation in performance (see the table below). This further confirms that the fidelity refinement stage makes an important contribution to the final generation quality.
>
> |Methods|GenEval↑|DPG↑|FID↓|KID×10²↓|IS↑|
> |-|-|-|-|-|-|
> |Infinity|0.73|83.12|26.64|1.26|42.61|
> |Infinity (w/o Fidelity Refinement Stage)|0.72|82.58|27.58|1.40|41.35|
> |Ours|0.72|82.86|26.91|1.36|42.18|
>
> **W2&Q2(2): It remains unclear whether the same negligible performance drop would hold at lower resolutions.**
>
> In this paper, we focus on applying our method to high-resolution image generation rather than to models that use a resolution of 256 as the final output, as such a resolution is too low to generate high-quality images with rich details. Instead, we apply our method at the intermediate scale corresponding to a 256 resolution in Infinity (i.e., scale 16), and observe only a 1.02× speedup. This is because lower scales require very little computation time (e.g., around 50ms in scale16). \
> To ensure the completeness of our experiments, we also combine our method with lower-resolution models (e.g., 256×256) for class-conditional generation on the ImageNet dataset. As shown in the table below, our method achieves comparable results while providing acceleration. A key challenge in accelerating 256x256 resolution models lies in their smaller latent dimensionality (i.e., 16), compared to 64 for higher-resolution images (i.e., 1024), which shows in weaker low-rank properties. Further exploration of the trade-off between acceleration and quality in 256x256 resolution models will be part of our future work.
>
> |Models|#Latency↓|FID↓|IS↑|Pre↑|Rec↑|
> |-|-|-|-|-|-|
> |VAR (d16)|0.48s|3.52|281.8|0.85|0.48|
> |Ours (d16)|0.4s|4.17|235.0|0.81|0.51|
> |VAR (d30)|2.2s|2.06|305.0|0.81|0.59|
> |Ours (d30)|1.8s|2.69|273.8|0.79|0.60|
>
> **Q3: Was the boundary of the fidelity refinement stage determined for STAR?**
>
> Existing acceleration methods typically decide the speedup stages based on heuristic approaches. In contrast, we propose a systematic analysis framework to determine which steps can be accelerated. Although our method does not automatically determine the acceleration stages, it is effective across diverse VAR architectures and consistently achieves meaningful speedups. To achieve optimal acceleration performance, the determination of stage boundaries could benefit from our analysis framework. As a consequence, our method does not rely on fixed or manually tuned boundaries, instead it provides a general way to determine them.  For example, in STAR, our analysis shows that the fidelity refinement stage emerges later than in Infinity, which leads to different choices of acceleration scales. We will include the analysis for STAR in the supplementary material. \
> In future work, we plan to apply our method to other VAR variants. For example, REPA [1] could be integrated to align the VAR feature during the training stages, which may alter the order of structural and semantic generation. In such cases, our analytical approach is still expected to remain applicable.
>
> [1] Representation Alignment for Generation: Training Diffusion Transformers is Easier than You Think

---

> > ### Author Rebuttal · Reviewer_7web · 2026-04-03
> >
> > Thank the authors for their thorough rebuttal. I am satisfied that Q1 and Q3 have been well addressed. Regarding Q2, I note that the performance degradation becomes more pronounced at 256×256 resolution, while the corresponding speedup gains are weakened. This suggests there remains room for further investigation into how to better navigate the trade-off between acceleration and output quality in practical settings. Overall, I remain positive about this work and will maintain my current score.

---

> > > ### Author Response · Authors · 2026-04-08
> > >
> > > We sincerely appreciate Reviewer 7web for their time, recognition of our thorough rebuttal, and positive evaluation of this work. We are pleased that our responses have addressed Q1 and Q3. Regarding Q2, we adopt the default setting of $\alpha$ = 0.96, achieving a 1.2$\times$ speedup. For 256$\times$256 resolution image generation, we further explore a more aggressive setting of $\alpha$ (i.e., 0.60). As shown in the table below, setting $\alpha$ = 0.60 achieves a 1.33$\times$-1.49$\times$ speedup while maintaining a comparable level of performance degradation, resulting in a better trade-off between acceleration and performance. In future work, we will continue investigating improved strategies to further optimize the balance between acceleration and output quality at this low resolution.
> > >
> > > | Models        | α    | Speed↑ | Latency↓ | FID↓ | IS↑   | Pre↑ | Rec↑ |
> > > |--------------|------|--------|----------|------|-------|------|------|
> > > | VAR (d16)    | -    | 1.0x   | 0.48s    | 3.52 | 281.8 | 0.85 | 0.48 |
> > > | Ours (d16)   | 0.96 | 1.2x   | 0.4s     | 4.17 | 235.0 | 0.81 | 0.51 |
> > > | Ours (d16)   | 0.60 | 1.33x  | 0.36s    | 4.34 | 228.5 | 0.81 | 0.50 |
> > > | VAR (d30)    | -    | 1.0x   | 2.2s     | 2.06 | 305.0 | 0.81 | 0.59 |
> > > | Ours (d30)   | 0.96 | 1.2x   | 1.8s     | 2.69 | 273.8 | 0.79 | 0.60 |
> > > | Ours (d30)   | 0.60 | 1.49x  | 1.47s    | 2.93 | 257.5 | 0.77 | 0.60 |

---

### Official Review · Reviewer_4DAk · 2026-03-12

**Soundness:** 3
**Presentation:** 3
**Significance:** 3
**Originality:** 3
**Overall Recommendation:** 4
**Confidence:** 4

**Summary:**

This paper proposes FasterVAR, a training-free acceleration method for visual autoregressive (VAR) models. The method divides the multi-scale generation process into semantic establishment, structure establishment, and fidelity refinement stages, and applies acceleration only in the late refinement stage while preserving the original inference process in earlier stages. It combines late-stage CFG removal with low-rank feature approximation using random projection and representative token restoration. Experiments on models such as Infinity and STAR show clear speedups with only minor quality degradation, suggesting that stage-aware approximation is an effective strategy for accelerating VAR inference.

**Compliance With Llm Reviewing Policy:**

Affirmed.

**Ethical Review Concerns:**

Overall, my main concerns have been resolved after rebuttal, and I maintain my score.

**Final Justification:**

The rebuttal satisfactorily addresses the main concerns raised in my original review. The authors clarified the practical determination of the boundary between structure establishment and fidelity refinement, provided a more understandable explanation of how projected tokens are aligned and restored in representative token restoration, and added hardware results showing that the speedup generalizes across devices. These additions improve both the technical clarity and the empirical completeness of the paper.

Given that my key concerns have been resolved, I view the paper more favorably after rebuttal. My final score is therefore maintained accordingly.

**Key Questions For Authors:**

1. How is the boundary between the structure establishment stage and the fidelity refinement stage determined in practice? Is it fixed for each model, or can it be selected automatically?

2. Could the authors clarify the connection between random projection and representative token restoration? In particular, how are projected tokens aligned with spatial tokens in practice?

3. Since the method reduces Transformer computation but adds projection and restoration steps, could the authors provide a clearer latency breakdown and comment on whether the speedup is stable across different hardware settings?

**Limitations:**

Yes

**Strengths And Weaknesses:**

The paper addresses a practical and relevant problem, namely accelerating visual autoregressive inference, and proposes a training-free, stage-aware method with clear engineering value. A key strength is the intuition of preserving the semantic and structure establishment stages while accelerating only the late fidelity refinement stage. The reported results on Infinity and STAR are promising and suggest that the method can achieve meaningful speedups with only minor quality degradation.

A main weakness is that some method details remain unclear. In particular, the boundary between stages appears somewhat heuristic, and it is not fully clear how well this choice transfers across different backbones or settings. In addition, the connection between random projection and representative token restoration is not entirely well specified, which makes the implementation harder to interpret and reproduce. Finally, since the method adds projection and restoration steps while reducing Transformer computation, it would be helpful to clarify how robust the reported speedup is across different hardware conditions.

---

> ### Author Rebuttal · Authors · 2026-03-31
>
> We sincerely thank the reviewer 4DAk for the thoughtful and constructive comments.
>
> We use **W** to abbreviate Weaknesses, and **Q** to represent Questions.
>
> **W1&Q1: How is the boundary between the structure establishment stage and the fidelity refinement stage determined in practice? Is it fixed for each model, or can it be selected automatically?**
>
> Existing acceleration methods typically decide the speedup stages based on heuristic approaches. In contrast, we propose a systematic analysis framework to determine which steps can be accelerated. Although our method does not automatically determine the acceleration stages, it is effective across diverse VAR architectures and consistently achieves meaningful speedups. To achieve optimal acceleration performance, the determination of stage boundaries could benefit from our analysis framework. As a consequence, our method does not rely on fixed or manually tuned boundaries, instead it provides a general way to determine them. \
> In future work, we plan to apply our method to other VAR variants. For example, REPA [1] could be integrated to align the VAR feature during the training stages, which may alter the order of structural and semantic generation. In such cases, our analytical approach is still expected to remain applicable.
>
> [1] Representation Alignment for Generation: Training Diffusion Transformers is Easier than You Think
>
> **W2&Q2: Could the authors clarify the connection between random projection and representative token restoration? In particular, how are projected tokens aligned with spatial tokens in practice?**
>
> We use random projection to map the original $M$ tokens into $r$ tokens, thereby compressing the number of tokens. Due to the Johnson-Lindenstrauss (JL) property [2] of random projection, the $r$ tokens essentially capture the core information of the original $M$ tokens. After the forward process, the $r$ tokens need to be restored back to $M$ tokens to match the input of the VAE decoder. To minimize processing time during restoration, we propose Representative Token Restoration, where we treat the $r$ tokens (projected tokens) as the most important ones (spatial tokens) based on [3] among the $M$ tokens and perform substitution.
>
> [2] Extensions of lipschitz mappings into a hilbert space \
> [3] Fast monte-carlo algorithms for finding low-rank approximations
>
> **W3&Q3: The latency in different hardware settings.**
>
> We tested our method on different hardware platforms and achieved over 3× acceleration in all cases, demonstrating that our method provides acceleration across various devices.
>
> | Methods  | RTX3090        | A100           | H200           |
> |----------|----------------|----------------|----------------|
> | Infinity | 2.23s          | 1.81s          | 1.50s          |
> | Ours     | 0.64s (3.4×)   | 0.60s (3.02×)  | 0.48s (3.12×)  |

---

> > ### Author Rebuttal · Reviewer_4DAk · 2026-04-02
> >
> > Thank you for the clear and constructive rebuttal. My main concerns have been adequately addressed.

---

> > > ### Author Response · Authors · 2026-04-02
> > >
> > > We sincerely appreciate the reviewer 4DAk again for their time and review.

---

### Official Review · Reviewer_ioUJ · 2026-03-12

**Soundness:** 3
**Presentation:** 3
**Significance:** 2
**Originality:** 3
**Overall Recommendation:** 4
**Confidence:** 4

**Summary:**

The paper proposes a training-free acceleration method for VAR models by exploiting the stage-wise characteristics of scale-wise generation and applying low-rank approximation to late-stage features.

**Compliance With Llm Reviewing Policy:**

Affirmed.

**Ethical Review Concerns:**

No need to ethical review.

**Final Justification:**

The paper is well motivated and presents a practical training-free acceleration method for VAR models, supported by intuitive analysis and competitive empirical results across multiple models and benchmarks. The proposed design is simple and effective, and the stage-wise analysis provides useful insights into VAR generation.

The rebuttal further strengthens the paper by addressing several of my concerns. In particular, the authors provided additional results averaged over multiple random seeds, which improve the reliability of the analysis in Fig. 6. Additional evaluations on HPS and PickScore also help demonstrate that the method preserves perceptual quality and human preference. While some questions regarding generalization of stage boundaries remain, the clarifications and additional evidence are sufficient to support the overall claims.

Overall, I find the work technically solid and practically useful, and I maintain my recommendation.

**Key Questions For Authors:**

See weaknesses. Providing clearer experimental analysis and evaluation would strengthen the paper.

**Limitations:**

Discussing and showcasing some bad cases under acceleration will help to further refine this work.

**Strengths And Weaknesses:**

**Strengths:**
- The motivation experiments are intuitive and well designed. The paper provides a systematic analysis of how different scales and CFG settings affect the generation process, supporting the observation that VAR generation proceeds through stages of semantic establishment, structural refinement, and fidelity enhancement.
- To avoid the high computational cost of SVD decomposition, the paper introduces a random projection strategy for low-rank approximation and combines it with a Representative Token Restoration mechanism to efficiently reconstruct features. These design choices are simple and effective.
- The proposed method is evaluated on multiple base models (Infinity-2B, Infinity-8B, and STAR) and across several benchmarks, providing a relatively thorough empirical evaluation.

**Weaknesses:**
- In Fig. 6, the analysis of different ranks may be affected by randomness in the GenEval metric. It would be more convincing to report averaged results across multiple random seeds. Additionally, the qualitative examples on the right side of Fig. 6 do not clearly illustrate noticeable differences between different rank settings.
- The evaluation mainly focuses on semantic and spatial alignment metrics (e.g., GenEval and DPG). Additional evaluations related to aesthetics or human preference (e.g., HPS or PickScore) could provide a more comprehensive assessment of generation quality.
- The boundary between the semantic, structure, and fidelity stages appears to be determined empirically. It remains unclear whether the same stage partition generalizes to other VAR architectures or requires tuning for different models.

---

> ### Author Rebuttal · Authors · 2026-03-31
>
> We sincerely thank the reviewer ioUJ for the thoughtful and constructive comments.
>
> We use **W** to abbreviate Weaknesses, **Q** to represent Questions, and **L** to represent Limitations.
>
> **W1: Fig. 6 (Left) would be more convincing if averaged results across multiple random seeds were reported. The qualitative examples in Fig. 6 (Right) do not clearly illustrate noticeable differences between different rank settings.**
>
> Fig. 6 (Left) shows that the GenEval metrics for different ranks are around 0.72, with the highest value at 17.6% rank ($\alpha$=0.96), indicating the randomness of our proposal. To further validate this, we repeated the tests multiple times with random seeds and reported the averaged results in the table below. The table shows that the GenEval results for different ranks consistently remain around 0.72, with the best results at the 17.6% rank. The averaged results follow the same trend. The qualitative examples in Fig. 6 (Right) show that the results for different ranks are very close to the vanilla result, with only minor differences in details, such as the 'mouth' of the bear.
> | GenEval| 59.5% | 34.4% | 26.1% | 21.1% | 17.6% | 14.9% | 6.9% | 3.2%  |
> |--------|--------------|--------------|--------------|--------------|--------------|--------------|-------------|-------------|
> | Random Seeds 1 | 0.71688      | 0.71596      | 0.72040      | 0.72570      | 0.72599      | 0.72495      | 0.72555     | 0.72447     |
> | Random Seeds 2 | 0.71563      | 0.71214      | 0.72054      | 0.72502      | 0.72744      | 0.72321      | 0.72463     | 0.72717     |
> | Random Seeds 3 | 0.71305      | 0.71212      | 0.71236      | 0.71588      | 0.72722      | 0.71907      | 0.71299     | 0.70928     |
> | Random Seeds 4 | 0.71217      | 0.71277      | 0.71352      | 0.71792      | 0.72733      | 0.71831      | 0.71084     | 0.70830     |
> | Random Seeds 5 | 0.71275      | 0.71417      | 0.71281      | 0.71917      | 0.72179      | 0.72154      | 0.71961     | 0.72010     |
> | **Avgrage**| **0.71410**  | **0.71343**  | **0.71593**  | **0.72074**  | **0.72595**  | **0.72142**  | **0.71872** | **0.71786** |
>
> **W2: Additional evaluations related to aesthetics or human preference (e.g., HPS or PickScore) could provide a more comprehensive assessment of generation quality.**
>
> Thank you for your suggestion. In our method, we evaluate performance in the same way as the vanilla VAR paper, using the GenEval and DPG benchmarks. Additionally, we have incorporated commonly used metrics such as FID, KID and IS, as shown in Table 4, to assess perceptual quality. Following your advice, we conducted further comparisons of our approach with baselines using the HPS and PickScore metrics. The results, presented in the table below, show that our method maintains model performance. These evaluation results, along with the experiments in Table 3 and Table 4, demonstrate that our approach preserves perceptual quality, aesthetics, and human preference.
> | Methods  | HPS ↑ | PickScore ↑ |
> |----------|-------|-------------|
> | Infinity | 30.48 | 23.1278     |
> | Ours     | 30.07 | 23.0080     |
> | STAR     | 29.16 | 21.5665     |
> | Ours     | 29.06 | 21.2852     |
>
> **W3: The boundary between the semantic, structure, and fidelity stages appears to be determined empirically. It remains unclear whether the same stage partition generalizes to other VAR architectures or requires tuning for different models.**
>
> Existing acceleration methods typically decide the speedup stages based on heuristic approaches. In contrast, we propose a systematic analysis framework to determine which steps can be accelerated. Although our method does not automatically determine the acceleration stages, it is effective across diverse VAR architectures and consistently achieves meaningful speedups. To achieve optimal acceleration performance, the determination of stage boundaries could benefit from our analysis framework. As a consequence, our method does not rely on fixed or manually tuned boundaries, instead it provides a general way to determine them. \
> In future work, we plan to apply our method to other VAR variants. For example, REPA [1] could be integrated to align the VAR feature during the training stages, which may alter the order of structural and semantic generation. In such cases, our analytical approach is still expected to remain applicable.
>
> [1] Representation Alignment for Generation: Training Diffusion Transformers is Easier than You Think
>
> **L1: Discussing and showcasing some bad cases under acceleration will help to further refine this work.**
>
> The vanilla model encounters bad cases when generating complex poses, objects, and text. Our proposed method accelerates the vanilla model while inheriting its limitations. However, the acceleration method would be slightly different from the vanilla model. For example, while generating fine textures (hair, skin, etc.), the original model’s quality can be preserved, but the structure may be slightly altered.

---

> > ### Author Rebuttal · Reviewer_ioUJ · 2026-04-03
> >
> > Thank you for the detailed rebuttal and additional experiments. The clarifications and new results addressed my concerns and further strengthened the empirical evidence. I maintain my positive recommendation.

---

### Official Review · Reviewer_Aayt · 2026-03-13

**Soundness:** 3
**Presentation:** 2
**Significance:** 3
**Originality:** 2
**Overall Recommendation:** 3
**Confidence:** 3

**Summary:**

This paper try to acclerate the inference of VAR-style image generations without training. The authors analyze the VAR inference process and divide it into three distinct stages: semantic establishment, structure establishment, and fidelity refinement. They observe that the early steps are critical for building structural and semantic consistency and must be preserved. However, the later steps primarily refine image fidelity and can be heavily optimized by utilzing the low-rank properties of the features.

The author experimented on Infinity and START (both are T2I VAR style models), and showed 1.7x-3.4x speed accleration without performance drop.

**Compliance With Llm Reviewing Policy:**

Affirmed.

**Key Questions For Authors:**

1. In the analysis part of the paper (section 3.2), I in general feels that such observation (structure and semantic first, then fine details) might be discussed by multiple previous papers, not only on VAR, but also on diffusion models. What are the major differences made by the author compared to previous observations? and I think the author should disscuss on this point.

2. I found that the achieved speed up is hard to understand. Can the author do some computations on FLOPS? Also, what is the configuration when you test the speed? which batch size and what are the resolutions?

**Limitations:**

No discussion of limitations.

**Strengths And Weaknesses:**

## Strength:

Significant Performance Gains: The framework achieves impressive acceleration metrics, demonstrating a 3.4x speedup on Infinity-2B, a 2.7x speedup on Infinity-8B, and a 1.74x speedup on STAR-1.7B while maintaining image quality, which is impressive.

Training-Free Integration: The paper proposes a plug-and-play module that accelerates existing VAR models without the need for computationally expensive retraining.

Robustness:  The proposed methods seems to be quite robust across prompt, and across hyper-parameters.


## Weakness
I found the major weakness lies in presentations.

Figure Clarity and Standalone Readability: The presentation of the methodology needs refinement. Figures 3 and 4 are currently difficult to interpret independently, as understanding the notations requires the reader to cross-reference multiple equations (e.g., Equations 1, 3, 4, 5, 7, and 8). Making these figures more self-contained with clearer captions and simplified notations would significantly improve the paper's readability.

Ablation Clarity Regarding CFG vs. Novel Designs: The paper should clarify earlier in the text how much of the speedup is attributed to bypassing Classifier-Free Guidance (CFG) versus the novel low-rank feature designs. While Table 5 indicates that applying the semantic irrelevance strategy (CFG=0) alone yields a 1.5x speedup, this crucial breakdown is not immediately obvious. Highlighting this distinction earlier would better contextualize the core technical contributions of the Random Projection and Representative Token Restoration methods

---

> ### Author Rebuttal · Authors · 2026-03-31
>
> We sincerely thank the reviewer Aayt for the thoughtful and constructive comments.
>
> We use **W** to abbreviate Weaknesses, and **Q** to represent Questions.
>
> **W1: Figure Clarity and Standalone Readability.**
>
> W1: Sorry for the inconvenience brought from Fig.3 and Fig.4, where we integrated substantial knowledge and information about VAR models referring to the VAR and AREdit  papers. To facilitate the understanding of this whole process, we also had the algorithm demonstration in the Section.B of supplementary material. We plan to include a more detailed figure with clearer captions and simplified notations in the supplementary materials to better illustrate our method and the algorithm pipeline.
>
> **W2: The paper should clarify earlier in the text how much of the speedup is attributed to bypassing CFG versus the novel low-rank feature designs.**
>
> Thank you for the suggestion to highlight this distinction earlier. We didn’t include the speedup ratio of the CFG mechanism earlier in the method section, where we focus on conceptual content. To alleviate your concerns, we will clarify the speedup attributed to bypassing the semantic irrelevance strategy (not directly a simple CFG=0) versus the novel low-rank feature designs at the end of Subsection “3.2. Observations, Semantic irrelevance at large-scale steps.” (Line 200).
>
> **Q1: What are the major differences made by the author compared to previous observations on diffusion models?**
>
> The major difference between diffusion models and VAR models lies in the semantic and structural information formation procedure, which arises from their architecture designs and generation mechanisms. The process of semantic and structural formation in diffusion models has been discussed in prior works [1,2,3]. By contrast, we pioneer the systematic analysis of the semantic and structural establishment in VAR models. Our analysis reveals that in VAR, semantics are determined earlier than the structure, whereas in diffusion models, structure is established before semantics. For example, in image editing tasks, P2P [1] performs semantic transformation through target text prompts modification (e.g., “cat” to “dog”), while preserving the structure by injecting the attention maps in early steps from the source branch into the target branch.
>
> [1] Prompt-to-prompt image editing with cross attention control. \
> [2] CleanDIFT: Diffusion Features without Noise. \
> [3] Generative modelling with inverse heat dissipation
>
> **Q2: Can the author do some computations on FLOPS? Also, what is the configuration when you test the speed? which batch size and what are the resolutions?**
>
> We use the GenEval benchmark to calculate the average inference time, with a batch size of 1 and the same 1024 resolution as the vanilla text-to-image VAR generation. Both the vanilla model and ours are executed on an RTX 3090, with a FLOPS of 35.7 TFLOPS. As shown in the table below, we also compute the average FLOPs using the GenEval benchmark. The results indicate that the FLOPs of our proposed method are significantly lower than those of the vanilla VAR models.
>
> | Methods  | FLOPS (RTX 3090) | FLOPs (G) |
> |------------------------------------------|-----------|--------------|
> | Infinity | 35.7 TFLOPS | 12170.61  |
> | Ours     | 35.7 TFLOPS | 7864.12|
> | STAR | 35.7 TFLOPS | 20910.63  |
> | Ours     | 35.7 TFLOPS | 12086.62 |

---

> > ### Author Rebuttal · Reviewer_Aayt · 2026-04-05
> >
> > My major concern about the FLOPs is fully resovled.

---

### Decision · Program_Chairs · 2026-04-30

**Decision:**

Accept (regular)

**Comment:**

Overall, the reviewers agree that this paper makes a strong contribution to accelerating visual autoregressive generation. The motivation is clear, the training-free and plug-in design is minimal, and it also shows consistent empirical speedups across multiple VAR backbones, with only minor quality degradation. Pre-rebuttal, there were some concerns, but the rebuttal addressed most of these points well. Reviewer consensus after rebuttal was mostly positive: the initially negative reviewer indicated that the main concerns were fully resolved, while the other reviewers also maintained positive scores. Therefore, I recommend acceptance.